# Flexible linkers in CaMKII control the balance between activating and inhibitory autophosphorylation

**Moitrayee Bhattacharyya[1,2,3†‡], Young Kwang Lee[1,2,4†§], Serena Muratcioglu[1,2,3], Baiyu Qiu[1,2,3], Priya Nyayapati[1,2,3], Howard Schulman[5], Jay T Groves[1,2,4,6]\*, John Kuriyan[1,2,3,4,6]\***

[1]Department of Molecular and Cell Biology, University of California, Berkeley, Berkeley, United States; [2]California Institute for Quantitative Biosciences (QB3), University of California, Berkeley, Berkeley, United States; [3]Howard Hughes Medical Institute, University of California, Berkeley, Berkeley, United States; [4]Department of Chemistry, University of California, Berkeley, Berkeley, United States; [5]Panorama Institute of Molecular Medicine, Sunnyvale, United States; [6]Physical Biosciences Division, Lawrence Berkeley National Laboratory, Berkeley, Berkeley, United States

**\*For correspondence:**
jtgroves@lbl.gov (JTG);
kuriyan@berkeley.edu (JK)

[†]These authors contributed equally to this work

**Present address:** [‡]Department of Pharmacology, Yale University, New Haven, United States; [§]Department of Chemistry and Biochemistry, San Diego State University, San Diego, United States

**Abstract** The many variants of human $Ca^{2+}$/calmodulin-dependent protein kinase II (CaMKII) differ in the lengths and sequences of disordered linkers connecting the kinase domains to the oligomeric hubs of the holoenzyme. CaMKII activity depends on the balance between activating and inhibitory autophosphorylation (on Thr 286 and Thr 305/306, respectively, in the human α isoform). Variation in the linkers could alter transphosphorylation rates within a holoenzyme and the balance of autophosphorylation outcomes. We show, using mammalian cell expression and a single-molecule assay, that the balance of autophosphorylation is flipped between CaMKII variants with longer and shorter linkers. For the principal isoforms in the brain, CaMKII-α, with a ~30 residue linker, readily acquires activating autophosphorylation, while CaMKII-β, with a ~200 residue linker, is biased towards inhibitory autophosphorylation. Our results show how the responsiveness of CaMKII holoenzymes to calcium signals can be tuned by varying the relative levels of isoforms with long and short linkers.

## Introduction

A characteristic feature of signaling by protein kinases is that the kinases are themselves regulated by phosphorylation. For most kinases, this involves the phosphorylation of one or more residues in the activation loop, a regulatory element located at the active site of the kinase (*Huse and Kuriyan, 2002*; *Johnson et al., 1996*; *Nolen et al., 2004*). An exception is provided by $Ca^{2+}$/calmodulin-dependent protein kinase II (CaMKII), an enzyme that has crucial roles in animal-cell signaling, particularly in the brain and in cardiac tissue (*Bayer and Schulman, 2019*; *Bhattacharyya et al., 2019*; *Griffith, 2004*; *Hudmon and Schulman, 2002*; *Kennedy, 2013*; *Lisman et al., 2012*; *Stratton et al., 2013*). CaMKII has no phosphorylation sites within its activation loop. Instead, the activity of CaMKII is modulated by autophosphorylation on two sites, Thr 286 and Thr 305/306 (residue numbering corresponds to the sequence of human CaMKII-α), both located in a regulatory segment that immediately follows the kinase domain (*Figure 1a,d*).

Phosphorylation of Thr 286 confers $Ca^{2+}$/calmodulin-independent activity, referred to as autonomy (*Lai et al., 1986*; *Lou et al., 1986*; *Miller et al., 1988*; *Miller and Kennedy, 1986*; *Saitoh and Schwartz, 1985*; *Schworer et al., 1986*). We refer to Thr 286 phosphorylation as activating phosphorylation, since it prolongs the active state. In contrast, phosphorylation on Thr 305/306 is

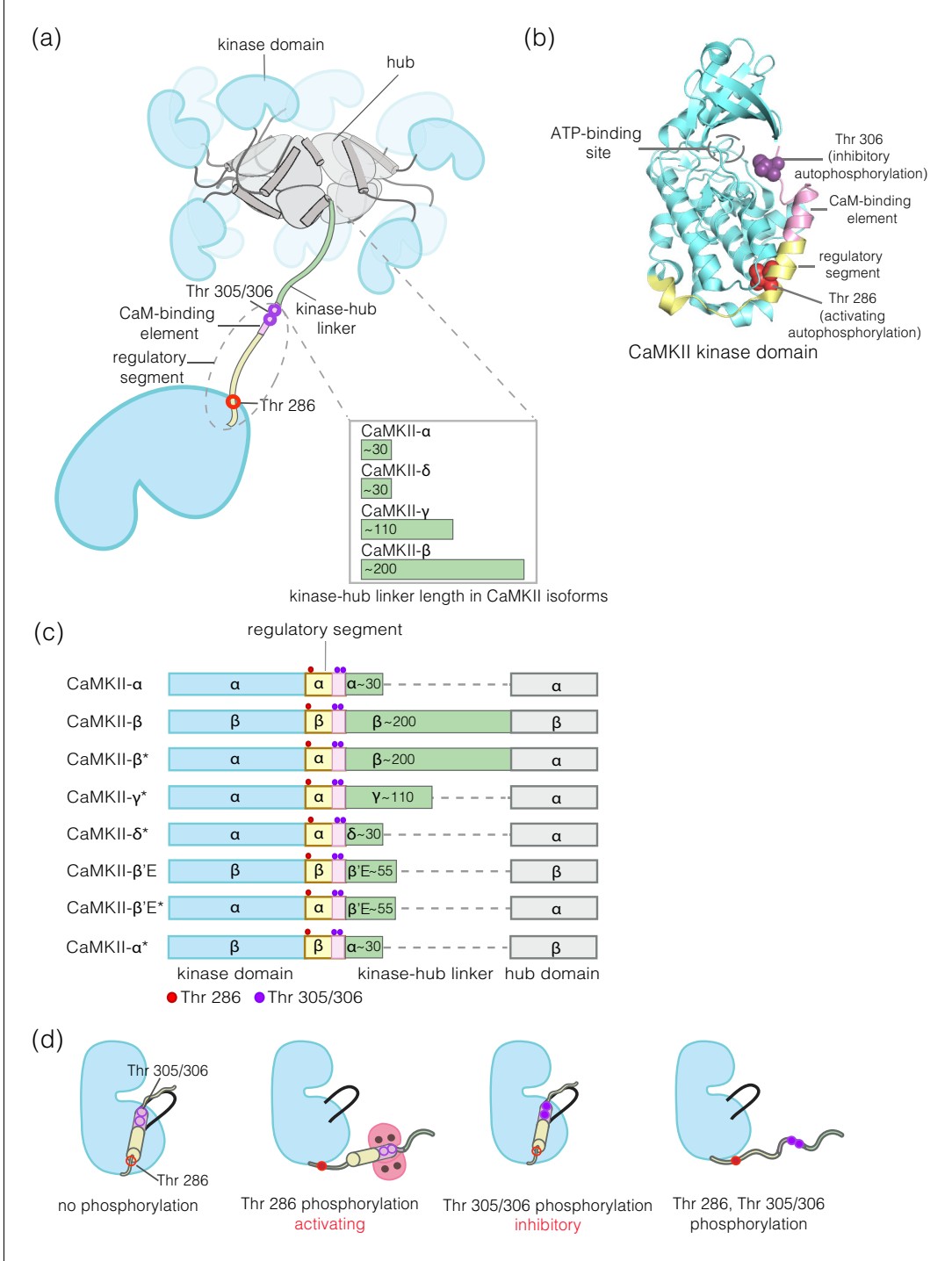

**Figure 1.** Structural organization and Ca$^{2+}$/CaM-dependent activation of CaMKII. (**a**) CaMKII is organized as a holoenzyme with kinase domains connected to a central dodecameric/tetradecameric hub by a regulatory segment and a flexible linker, referred to as the kinase-hub linker. All the domains are labeled and this color scheme used will be maintained throughout. The kinase-hub linker is the principle difference between the four CaMKII isoforms: α/β/γ/δ. (**b**) Crystal structure of the autoinhibited kinase domain from human CaMKII-δ (PDB ID: 2VN9) (*Rellos et al., 2010*). The regulatory segment places Thr 306 optimally for *cis*-phosphorylation, while Thr 286, at the base of the kinase can only be phosphorylated in trans. (**c**) Schematic diagram showing the design principle for all the constructs used in this study. (**d**) Depiction of the four possible phosphorylation states of CaMKII.

The online version of this article includes the following figure supplement(s) for figure 1:

**Figure supplement 1.** Amino acid sequences for the kinase-hub linker in the four isoforms of human CaMKII.

inhibitory, because it blocks the binding of $Ca^{2+}$/CaM (*Colbran, 1993*; *Hanson and Schulman, 1992*; *Hashimoto et al., 1987*; *Lou and Schulman, 1989*; *Patton et al., 1990*). Mutation of these autophosphorylation sites results in profound alterations in learning and memory (*Elgersma et al., 2002*; *Giese et al., 1998*; *Giese and Mizuno, 2013*; *Küry et al., 2017*; *Silva et al., 1992*).

CaMKII is organized into a dodecameric or tetradecameric holoenzyme consisting of a central ring-shaped hub assembly, to which the kinase domains are connected by flexible linkers (the kinase-hub linkers) (*Figure 1a*; *Chao et al., 2011*; *Myers et al., 2017*; *Rosenberg et al., 2005*). The four isoforms of human CaMKII, denoted α, β, γ, and δ, and their ~ 40 splice variants, differ principally in the lengths and sequences of the kinase-hub linkers. The kinase and the hub domains of these isoforms share about 90% and 80% sequence identity, respectively (*Tombes et al., 2003*). The kinase-hub linkers are highly conserved for a given CaMKII isoform, indicating that there are important isoform-specific roles for these linkers, but what these are is not understood. CaMKII-α and CaMKII-δ have linkers spanning 31 residues each, while CaMKII-β and CaMKII-γ have longer linkers (218 and 110 residues, respectively, in their principal variants). The sequences of the CaMKII kinase-hub linkers are consistent with their being intrinsically disordered, leading to dynamic variability in the disposition of the kinase domains with respect to the central hub (*Myers et al., 2017*).

Our current understanding of the autoregulation of CaMKII is based on studies that have focused primarily on CaMKII-α, and it has been assumed that the other three isoforms function similarly. The regulatory segment of CaMKII blocks the kinase active site in the basal state, and this inhibition is released by the binding of $Ca^{2+}$/calmodulin ($Ca^{2+}$/CaM) to the regulatory segment (*Figure 1d*). The binding of $Ca^{2+}$/CaM facilitates the *trans*-autophosphorylation of Thr 286 within the regulatory segment of one CaMKII subunit by the activated kinase domain of another subunit in the same holoenzyme (*Hanson et al., 1994*; *Rich and Schulman, 1998*). How the balance of phosphorylation between the activating and inhibitory sites is controlled in the different isoforms of CaMKII is poorly understood.

We developed a single-molecule assay to measure the phosphorylation status of CaMKII expressed in mammalian cells. In this assay, CaMKII is fused to an N-terminal fluorescent protein (a monomeric variant of enhanced GFP, mEGFP, *Cormack et al., 1996*; *Zacharias et al., 2002*) and to a biotin tag, enabling the capture and visualization, by total internal reflection fluorescence (TIRF) microscopy, of individual CaMKII holoenzymes on a glass slide coated with streptavidin (*Figure 2a*). Since CaMKII is captured directly from the cell lysate, this procedure avoids difficulties in expressing and purifying various isoforms of CaMKII, particularly those with longer linkers. In addition, the use of a flow cell allows CaMKII holoenzymes to be activated under different reaction conditions after immobilization on glass. Site-specific antibodies that recognize phosphorylated Thr 286 (pThr 286) or phosphorylated Thr 305/Thr 306 (pThr 305/306) can be used to probe the extent of phosphorylation on activating and inhibitory sites on individual holoenzymes (*Figure 2a*).

Using this assay, we discovered that activation of CaMKII-α, which has a 31-residue kinase-hub linker, results in robust autophosphorylation on the activating site (Thr 286), with relatively little inhibitory phosphorylation on Thr 305/306. This is consistent with expectation (*Baucum et al., 2015*). Unexpectedly, CaMKII-β, with a longer ~200 residue linker, undergoes robust inhibitory autophosphorylation on Thr 305/306, with less activating phosphorylation on Thr 286. We also monitored autophosphorylation for an artificial construct of CaMKII that is identical to CaMKII-α, except that the kinase-hub linker is replaced by a 218-residue linker from CaMKII-β (*Figure 1c*). The autophosphorylation pattern for this construct resembles that of CaMKII-β, showing that the difference in activating and inhibitory autophosphorylation between the two isoforms is due entirely to differences in the kinase-hub linkers.

When CaMKII is inactivated after activation, by removing $Ca^{2+}$/CaM and ATP, we found that the dephosphorylation by phosphatases of activating phosphorylation at Thr 286 is much slower than dephosphorylation of inhibitory phosphorylation at Thr 305/306, which is fast. The protection of activating phosphorylation and the rapid reversal of inhibitory phosphorylation by phosphatases allows CaMKII to act as an integrator of $Ca^{2+}$ pulses in cells. CaMKII-α and CaMKII-β can assemble to form mixed holoenzymes, with the balance between activating and inhibitory autophosphorylation determined by the ratio of the two isoforms. Thus, the calcium responsiveness of CaMKII holoenzymes can be tuned by varying the relative levels of the two isoforms.

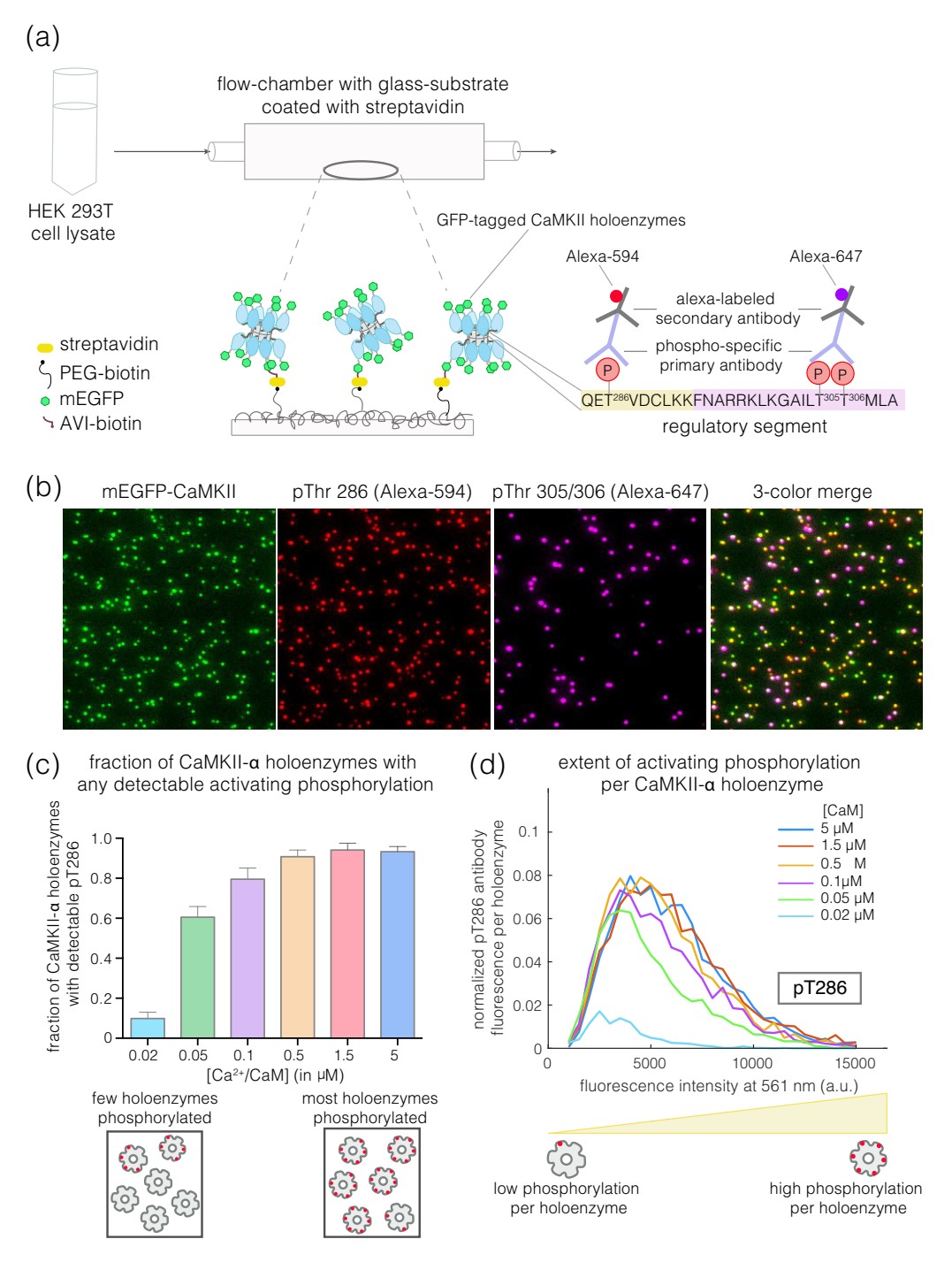

**Figure 2.** Mammalian expression-based single-molecule Total Internal Reflection Fluorescence (TIRF) assay. (a) Schematic diagram showing the experimental setup. Biotinylated mEGFP-CaMKII overexpressed in HEK 293T cells was pulled down directly from diluted cell lysate, allowing visualization at a single-molecule resolution. The immobilization onto glass substrates functionalized with streptavidin relies on the interaction between biotinylated CaMKII and streptavidin. Autophosphorylation status of CaMKII holoenzymes can be measured using phosphospecific primary antibodies and Alexa-labeled secondary antibodies. (b) Representative single-molecule TIRF images showing mEGFP-CaMKII holoenzymes (green dots), phosphorylation at Thr 286 (red dots) and phosphorylation at Thr 305/306 (purple dots) from left to right. A 3-color merge of these images reports on the fraction of CaMKII holoenzymes that are phosphorylated at Thr 286 and/or Thr 305/306. (c) Fraction of CaMKII-α that shows detectable phosphorylation at Thr 286 is plotted for different Ca$^{2+}$/CaM concentrations ranging from 0.02 µM to 5 µM. The cartoon at the bottom depicts two extreme cases, where only a few holoenzymes are phosphorylated or where most holoenzymes are phosphorylated. (d) Distribution of intensity for pThr 286 (561 nm), at different Ca$^{2+}$/CaM concentrations, for CaMKII-α holoenzymes with detectable phosphorylation (see Materials and methods for details

*Figure 2 continued on next page*

*Figure 2 continued*

of normalization). The cartoon at the bottom shows that a right-shift in the peak value of the intensity histogram represents a higher extent of phosphorylation within a CaMKII holoenzyme.

The online version of this article includes the following figure supplement(s) for figure 2:

**Figure supplement 1.** Validation of phosphospecific antibodies.

## Results and discussion

### A single-molecule assay for measuring CaMKII autophosphorylation

We designed a single-molecule assay to measure autophosphorylation of CaMKII, relying on immediate capture of the holoenzyme after lysis of mammalian cells overexpressing CaMKII (see Materials and methods) (*Figure 2a*). This procedure is based on a general single-molecule immunofluorescence assay reported previously (*Jain et al., 2011*). Our assay enables the rapid isolation of CaMKII for single-molecule analysis while minimizing heterogeneity that can result from proteolysis or aggregation during conventional purification. In this assay, GFP-CaMKII holoenzymes, whether or not phosphorylated, are detected with 488 nm laser excitation (green channel). The phosphorylation levels on Thr 286 and Thr 305/306 are monitored using phosphospecific antibodies labeled with Alexa-594 or Alexa-647, detected with 561 nm (red channel) or 640 nm laser excitation (purple channel), respectively (*Figure 2a*); see Materials and methods. Representative three-color TIRF images from these experiments are shown in *Figure 2b*.

We quantify the extent of phosphorylation in two ways. First, we measure the fraction of CaMKII holoenzymes that show detectable phosphorylation at Thr 286, regardless of the extent of phosphorylation, by monitoring the co-localization of green and red spots (*Figure 2c*). To quantify the extent of activating phosphorylation per holoenzyme, we compile intensity distributions for pThr 286 (red channel), taking into account both holoenzymes that are phosphorylated and those that show no detectable phosphorylation (*Figure 2d*, see Materials and methods for the details of normalization). The integrated intensity of the pThr 286 signal, normalized by the total number of holoenzymes detected (i.e., the number of green spots), reflects the mean level of phosphorylation per holoenzyme (*Figure 2d*). Phosphorylation at Thr 305/306 is quantified similarly, using the appropriate antibody.

We used this assay to measure autophosphorylation levels for CaMKII upon activation. Human CaMKII-α overexpressed in HEK 293T cells and captured from cell lysate is in an inactive state, with no detectable phosphorylation at Thr 286 or Thr 305/306 (data not shown). We activated CaMKII-α by flowing $Ca^{2+}$/CaM and ATP over the glass coverslip to which the captured holoenzymes are tethered (*Figure 2a*). Increasing the concentration of $Ca^{2+}$/CaM from 20 nM to 5 μM, while maintaining a constant saturating concentration of $Ca^{2+}$ (100 μM), resulted in the detection of increasing levels of phosphorylation at Thr 286 (*Figure 2c–d*). The $EC_{50}$ value for CaMKII-α activation by $Ca^{2+}$/CaM derived from these measurements is in the range of 50–100 nM. This is consistent with previously reported values for the $EC_{50}$ of CaMKII-α activation by $Ca^{2+}$/CaM (*Chao et al., 2011*; *Schulman, 1984*; *Sloutsky et al., 2019*).

A limitation of this assay is that the integrated intensity values cannot be used to determine the absolute number of phosphate groups within each holoenzyme, because we do not know whether every phosphate group necessarily has an antibody bound to it when the maximum signal is obtained. Detection of phosphorylation may be less than complete because of steric interference between antibodies, or interference with the glass support. Due to this limitation, we use the values of the integrated antibody fluorescence intensities to provide a relative, rather than an absolute, measure of the extent of phosphorylation.

### CaMKII-β, with a long kinase-hub linker, acquires inhibitory autophosphorylation more readily than CaMKII-α

We used the single-molecule assay to compare the levels of phosphorylation at the activating (Thr 286) and inhibitory (Thr 305/306) sites in CaMKII-α and CaMKII-β. We expressed CaMKII-α and CaMKII-β separately in HEK 293T cells and, in parallel experiments, captured the CaMKII isoforms on glass, followed by activation with high concentrations of $Ca^{2+}$/CaM (5 μM) and 500 μM ATP.

These experiments reveal clear differences in the extent of autophosphorylation at the inhibitory sites in CaMKII-α and CaMKII-β. Whereas 95% of the CaMKII-β holoenzymes show detectable phosphorylation at Thr 305/306, only 30% of the CaMKII-α holoenzymes do so (*Figure 3a*-right, inset). The mean value of the integrated intensity for Thr 305/306 phosphorylation per holoenzyme is 5-fold higher for CaMKII-β than for CaMKII-α (*Figure 3a*-right). The α and β isoforms also differ in the extent of activating phosphorylation on Thr 286, with CaMKII-α showing more Thr 286 phosphorylation than CaMKII-β (*Figure 3a*-left).

We verified that the reduced levels of pThr 305/306 seen for CaMKII-α are not due to an intrinsic inability to phosphorylate this site. We first activated CaMKII-α on glass, as before, and then washed away the $Ca^{2+}$/CaM. This was followed by the addition of ATP to the flow cell. ATP treatment of pre-activated CaMKII led to robust phosphorylation of CaMKII-α on Thr 305/306, at levels comparable to that for CaMKII-β (*Figure 3—figure supplement 1 (a-c)*).

We found that the differences in autophosphorylation between the α and β isoforms are due entirely to the differences in the kinase-hub linkers. We created an artificial CaMKII construct in which the kinase-hub linker in CaMKII-α is replaced by that from CaMKII-β (we refer to this construct as CaMKII-β*, because the most distinctive difference between the two isoforms is the nature of the kinase-hub linker, *Figure 1c*). The autophosphorylation pattern for CaMKII-β* is essentially the same as for CaMKII-β. Each CaMKII-β* holoenzyme exhibits about 5-fold greater phosphorylation of Thr 305/306 than is the case for CaMKII-α (*Figure 3b*-right), with about 95% of the CaMKII-β* holoenzymes showing detectable phosphorylation at pThr 305/306. CaMKII-β* also shows a lower degree of activating phosphorylation on Thr 286 when compared to CaMKII-α (*Figure 3b*-left).

We use CaMKII-β* as a surrogate for CaMKII-β in most of the experiments reported in this paper. The sequences flanking the phosphosites being probed by the antibodies are the same in CaMKII-β* and CaMKII-α, whereas there are a few small differences in sequence between CaMKII-β and CaMKII-α in these regions. Thus, when comparing CaMKII-β* to CaMKII-α, observed differences in autophosphorylation are not due to differences in the affinities of the antibodies for their cognate sites.

We also tested two other isoforms, CaMKII-γ (110-residue kinase-hub linker) and CaMKII-δ (31-residue kinase-hub linker). We created surrogates for the two isoforms using CaMKII-α, denoted as CaMKII-γ* and CaMKII-δ*. As for CaMKII-β*, the kinase-hub linker in CaMKII-α is replaced with those from the γ and δ isoforms, respectively, in these surrogates. The results for CaMKII-γ*, with the long linker, resemble those for CaMKII-β/CaMKII-β* (*Figure 3—figure supplement 2*). Likewise, the results for CaMKII-δ*, with the short linker, resemble those for CaMKII-α (*Figure 3—figure supplement 2*). The sequences of the linkers in the four CaMKII isoforms are very different, and so these results suggest that linker length alone is an important determinant of phosphorylation outcome (*Figure 1—figure supplement 1*).

This conclusion was further reinforced by studying a naturally occurring splice-variant of CaMKII-β (CaMKII-β'E) that has a 55-residue linker, created by deletion of 155 residues. We also designed a construct where this truncated linker from CaMKII-β'E replaced the linker in CaMKII-α (CaMKII-β'E*). For both these constructs, we see a decrease in inhibitory phosphorylation and a small increase in activating phosphorylation when compared to CaMKII-β (*Figure 3—figure supplement 3a–b*). As an additional control, we designed a construct of CaMKII-β in which the kinase-hub linker was replaced by that of CaMKII-α (CaMKII-α*). The autophosphorylation outcome for CaMKII-α* is similar to that for CaMKII-α (*Figure 3—figure supplement 3c*).

CaMKII-α and CaMKII-β can assemble into mixed holoenzymes containing both isoforms (*Bennett et al., 1983*; *Brocke et al., 1999*; *Thiagarajan et al., 2002*). We co-transfected mEGFP-CaMKII-α and mCherry-CaMKII-β* in HEK 293T cells in four different ratios (1:0, 3:1, 1:1, and 0:1, α:β*, respectively) and activated the resultant holoenzymes after immobilization on glass (*Figure 4a*). The co-transfection of mEGFP-CaMKII-α and mCherry-CaMKII-β* resulted in predominantly heterooligomeric holoenzymes, with ~80% colocalization of red and green spots (data not shown). As the proportion of CaMKII-β* increases, there is an increase in the extent of Thr 305/306 phosphorylation (*Figure 4b–c*). Thus, alterations in the relative levels of CaMKII-α and CaMKII-β can result in the generation of mixed CaMKII holoenzymes with different responsiveness to $Ca^{2+}$/calmodulin.

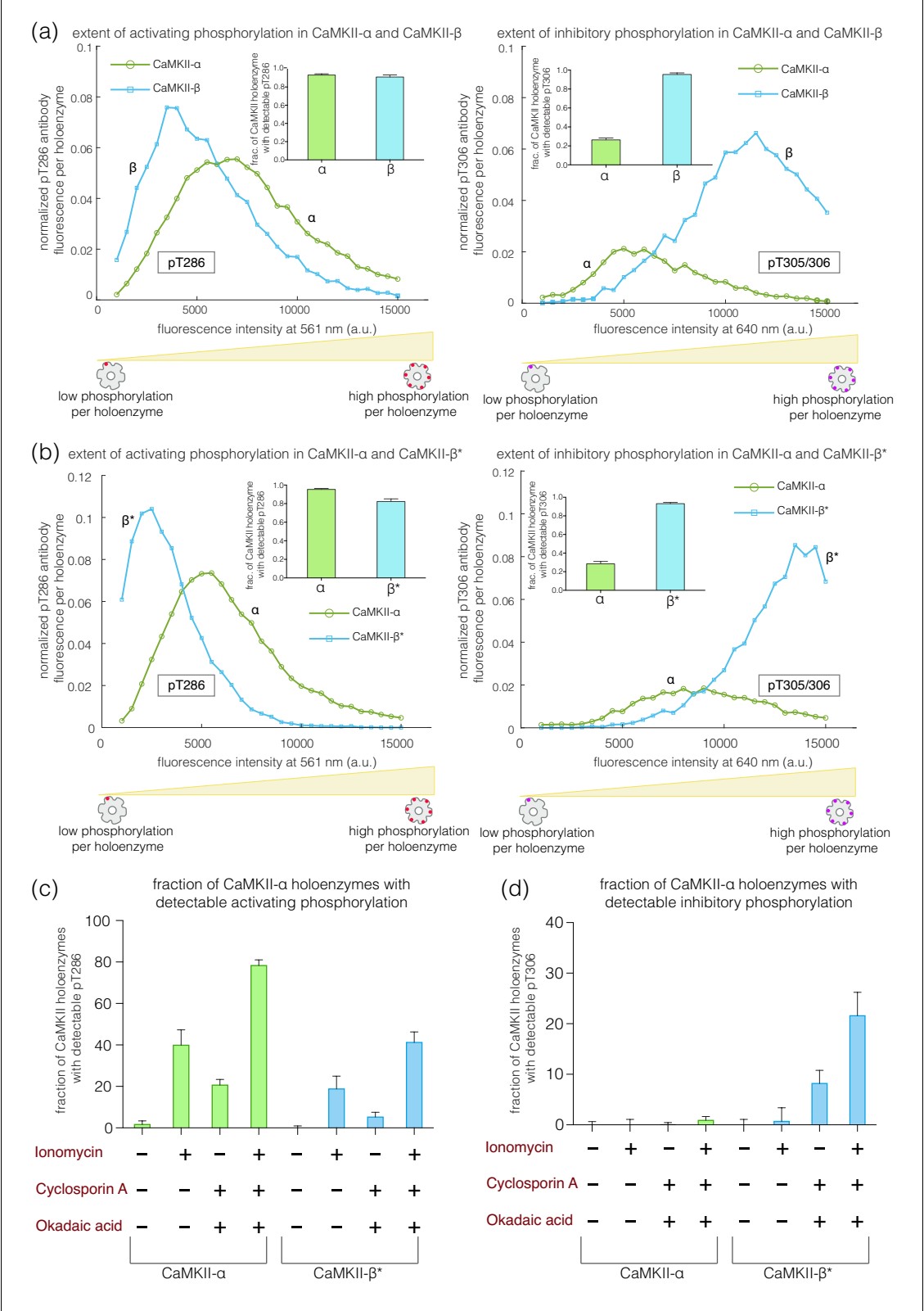

**Figure 3.** Autophosphorylation status of activated CaMKII-α and CaMKII-β/β*. (**a**) Comparison of the extent of autophosphorylation (intensity histogram) within CaMKII-α and CaMKII-β holoenzymes at the activating site (Thr 286) (left panel) and the inhibitory site (Thr 305/306) (right panel). The insets show the fraction of holoenzymes that exhibit any detectable phosphorylation for the corresponding phosphosite in CaMKII-α and CaMKII-β. (**b**) Comparison of the extent of autophosphorylation (intensity histograms) between CaMKII-α and CaMKII-β* holoenzymes at the activating site (Thr 286)

*Figure 3 continued on next page*

*Figure 3 continued*

(left panel) and the inhibitory site (Thr 305/306) (right panel). The insets show the fraction of holoenzymes that exhibit any detectable phosphorylation for the corresponding phosphosite in CaMKII-α and CaMKII-β* (see Materials and methods for details of normalization). (c) Autophosphorylation status of CaMKII after activation in HEK 293T cells using ionomycin. Fractions of CaMKII-α and CaMKII-β* that show detectable phosphorylation at Thr 286 are plotted for different conditions. (d) Fractions of CaMKII-α and CaMKII-β* that show detectable phosphorylation at Thr 305/306 are plotted for different conditions. + / - depicts the presence or absence of ionomycin and/or phosphatase inhibitors.

The online version of this article includes the following figure supplement(s) for figure 3:

**Figure supplement 1.** Inhibitory autophosphorylation status of activated CaMKII-α.

**Figure supplement 2.** Autophosphorylation status of activated CaMKII-γ* and CaMKII-δ*.

**Figure supplement 3.** Autophosphorylation status of activated CaMKII-β/β* variants.

## The altered balance of phosphorylation in CaMKII-α and CaMKII-β may arise due to differences in the rates of *cis* and *trans* autophosphorylation in the two isoforms

To understand how changes in the length of the kinase-hub linker can alter the balance of autophosphorylation, we analyzed a simple kinetic model for CaMKII activation in which the holoenzyme contains only two kinase domains (see Appendix). A kinetic model for the activation of a dodecameric CaMKII holoenzyme requires the specification of an extremely large number of intermediate states, and we have not pursued this. In our simple kinetic model, we assume that Thr 286 can only be phosphorylated in trans, because it is located too far from the active site of the kinase (*Figure 1b*,

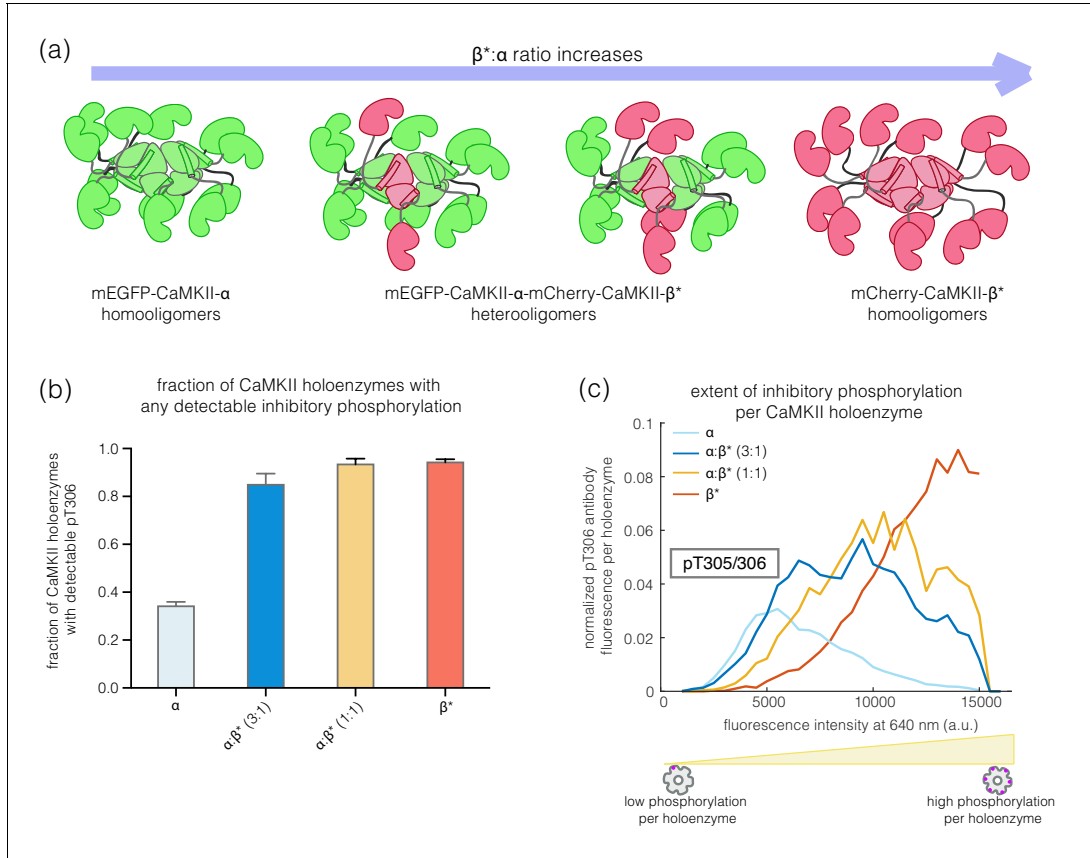

**Figure 4.** Inhibitory autophosphorylation in CaMKII-α-CaMKII-β* heterooligomers. (a) Schematic diagram showing that co-expression of GFP-CaMKII-α and mCherry-CaMKII-β* leads to the formation of heterooligomers. (b) Bar graph showing the fraction of holoenzymes that show detectable phosphorylation at the inhibitory site (Thr 305/306), which increases as the ratio of CaMKII-β* increases. (c) Intensity histogram for the homooligomers and heterooligomers. As the ratio of CaMKII-β* increases, there is a right-shift in the peak value of the intensity histogram (see Materials and methods for details of normalization).

and see *Figure 5*). We assume that Thr 305/306 can either be phosphorylated in cis, as suggested by crystal structures (*Rellos et al., 2010*), or in trans.

For *trans*-phosphorylation of either Thr 286 or Thr 305/306, the kinase acting as the enzyme has to have $Ca^{2+}$/CaM bound to it, otherwise the active site of the enzyme will be blocked. For *trans*-phosphorylation of Thr 286, the kinase acting as the substrate must also have $Ca^{2+}$/CaM bound to it, otherwise the Thr 286 is not accessible for phosphorylation (*Rich and Schulman, 1998*). Autophosphorylation of Thr 305/306 can only occur in the absence of $Ca^{2+}$/CaM binding to the substrate kinase, because otherwise Thr 305 and Thr 306 would be covered by $Ca^{2+}$/CaM. A key idea behind this model is that we assume that the longer linker in CaMKII-β slows down the rates of *trans*-phosphorylation reactions within a holoenzyme when compared to CaMKII-α, but that the rates of *cis*-phosphorylation are the same in the two isoforms (*Sørensen and Kjaergaard, 2019*).

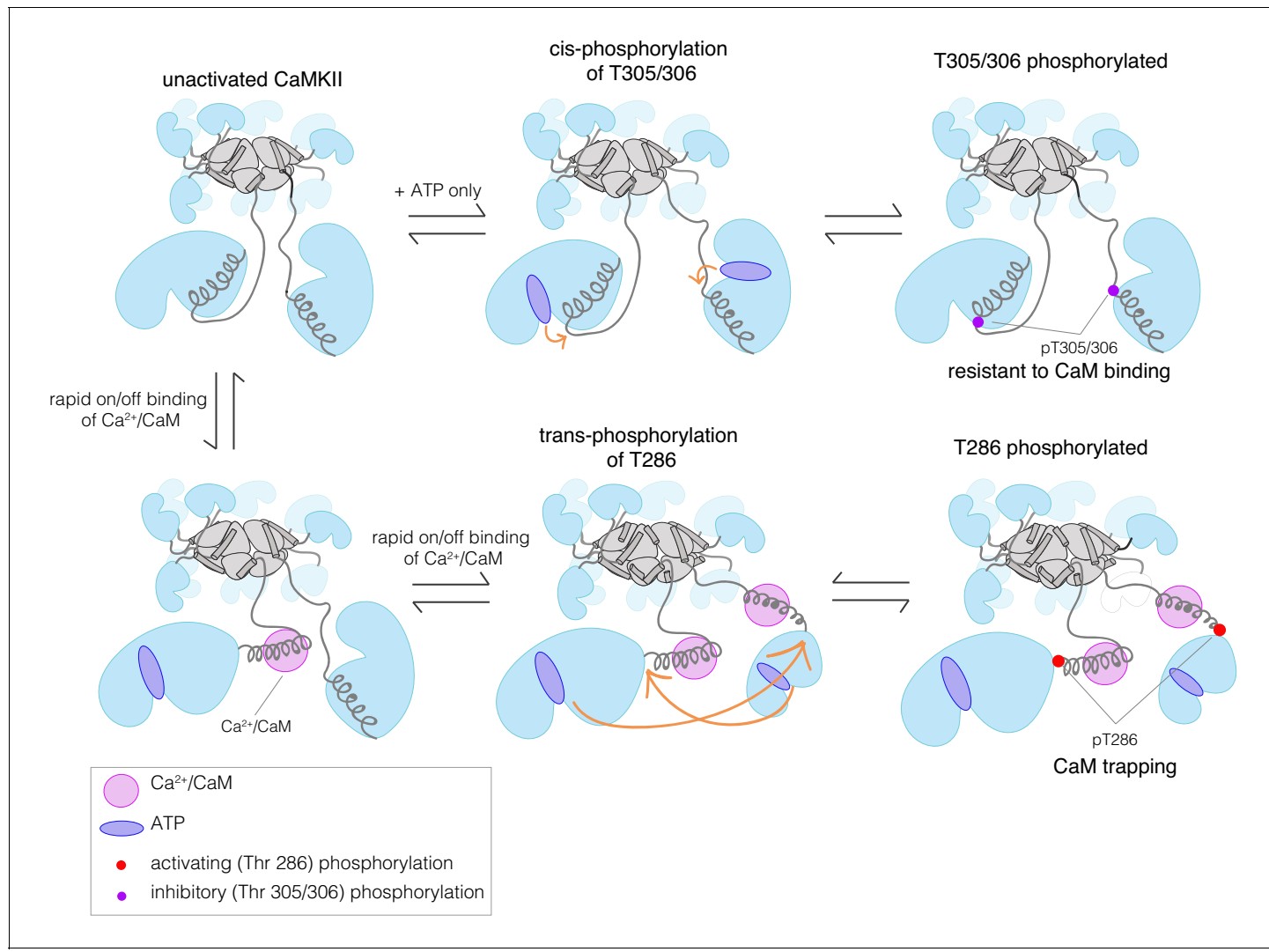

**Figure 5.** A simplified schematic diagram showing the key pathways for autophosphorylation at the activating and inhibitory sites, in the absence or presence of $Ca^{2+}$/CaM. While Thr 305/306 can get phosphorylated both in cis in the absence of $Ca^{2+}$/CaM or in trans in the presence of $Ca^{2+}$/CaM, autophosphorylation of Thr 286 can only happen in trans in the presence of $Ca^{2+}$/CaM. $Ca^{2+}$/CaM shows a rapid association and dissociation until CaMKII gets phosphorylated at Thr 286, when its affinity for $Ca^{2+}$/CaM increases by about 1000-fold. A detailed description of all the different reactions and conditions that form the basis of our kinetic model is provided in the Appendix.

The online version of this article includes the following figure supplement(s) for figure 5:

**Figure supplement 1.** Results from simulations of a simple kinetic model for autophosphorylation in CaMKII using Berkeley Madonna.

We analyzed the predicted autophosphorylation kinetics for the CaMKII dimer model using reasonable estimates for the rate constants in the model (see Appendix). Our results show that when the rate of *trans*-phosphorylation decreases relative to the rate of *cis*-phosphorylation, as is expected to occur with longer kinase-hub linkers, then there is a switch from Thr 286 phosphorylation dominating to Thr 305/306 phosphorylation dominating (*Figure 5—figure supplement 1a–b*, see Appendix). If Thr 286 is phosphorylated, the affinity for $Ca^{2+}$/CaM increases ~1000 fold, a phenomenon referred to as the calmodulin-trapping (*Meyer et al., 1992*; *Tse et al., 2007*). Since $Ca^{2+}$/CaM covers the inhibitory site (Thr 305/306), phosphorylation at this site is suppressed (*Figure 5*). The situation is reversed when the linker is long, and *cis*-phosphorylation is faster than *trans*-phosphorylation (*Figure 5—figure supplement 1b*). In this case, activating phosphorylation is suppressed, because $Ca^{2+}$/CaM binding is blocked.

## The balance between inhibitory and activating autophosphorylation is maintained upon activation of CaMKII in HEK 293T cells

We tested whether the balance between activating and inhibitory autophosphorylation is maintained when the enzyme is activated in cells. We overexpressed CaMKII-α or CaMKII-β* in HEK 293T cells, along with calmodulin, followed by ionomycin treatment to increase intracellular $Ca^{2+}$ levels. The cells were then lysed and CaMKII holoenzymes were captured and assayed as before, without any further treatment.

Phosphorylation at Thr 286 increases upon treatment with 10 µM ionomycin for both CaMKII-α and CaMKII-β*, as expected (*Baucum et al., 2015*; *Figure 3c*). CaMKII-α shows higher phosphorylation at Thr 286 as compared to CaMKII-β*, in agreement with our in vitro results (*Figure 3c*). However, no phosphorylation at Thr 305/306 is detected for either CaMKII-α or CaMKII-β* in this experiment (*Figure 3d*), which is consistent with previous reports (*Baucum et al., 2015*). There is depletion of ATP levels upon cell lysis, which could result in reduced kinase activity in the presence of phosphatase activity. We therefore incubated cells with okadaic acid and cyclosporin A (250 nM each), which are inhibitors of PP1 and calcineurin, respectively, for 45–60 min before activation. This treatment led to detectable levels of phosphorylation at Thr 305/306 for ~30% of the CaMKII-β* holoenzymes, but not for CaMKII-α (*Figure 3d*). This is consistent with the increased levels of Thr 305/306 phosphorylation seen for CaMKII-β* using the single-molecule assay.

## Activating autophosphorylation at Thr 286 is relatively resistant to dephosphorylation

We incubated CaMKII-α and CaMKII-β* with $Ca^{2+}$/CaM and ATP on glass, while including increasing amounts of λ-phosphatase, a potent non-specific phosphatase (*Zhuo et al., 1993*), in the activation buffer (*Figure 6a*). Active CaMKII-α is efficient at maintaining phosphorylation on Thr 286 in the face of competing phosphatase activity, while CaMKII-β* is less so (*Figure 6b,d*). For CaMKII-β*, high levels of inhibitory phosphorylation at Thr 305/306 are maintained in the presence of phosphatase (*Figure 6c,e*). In contrast, phosphorylation at Thr 305/306 is completely removed by the phosphatase for CaMKII-α (*Figure 6c,e*). Thus, the conclusion that CaMKII-α and CaMKII-β/β* more readily gain activating and inhibitory phosphorylation, respectively, holds true in the presence of phosphatases during activation.

We next investigated how efficiently the phosphatase can dephosphorylate the activating and inhibitory sites when the kinase activity is switched off, as would happen after a calcium pulse. We first activated CaMKII on glass, and then washed away $Ca^{2+}$/CaM and ATP (see Materials and methods, *Figure 7a*). In the previous experiments, we determined the concentration of phosphatase that results in the maximum reduction of phosphorylation levels during activation – we refer to this concentration of phosphatase as a 'saturating' concentration. With the kinase activity switched off, we initiated dephosphorylation reactions by adding a saturating concentration of λ-phosphatase to the phosphorylated CaMKII. The dephosphorylation reaction was then stopped by washing away the phosphatase at defined time points. The extent of phosphorylation at Thr 286 and Thr 305/306 was then measured as before (*Figure 7a*).

In the absence of competing kinase activity, pThr 305/306 is rapidly dephosphorylated for both CaMKII-α and CaMKII-β* (*Figure 7b*, *Figure 7—figure supplement 1a*). In contrast, the rate of dephosphorylation of pThr 286 is slow, with complete dephosphorylation taking more than 30 min

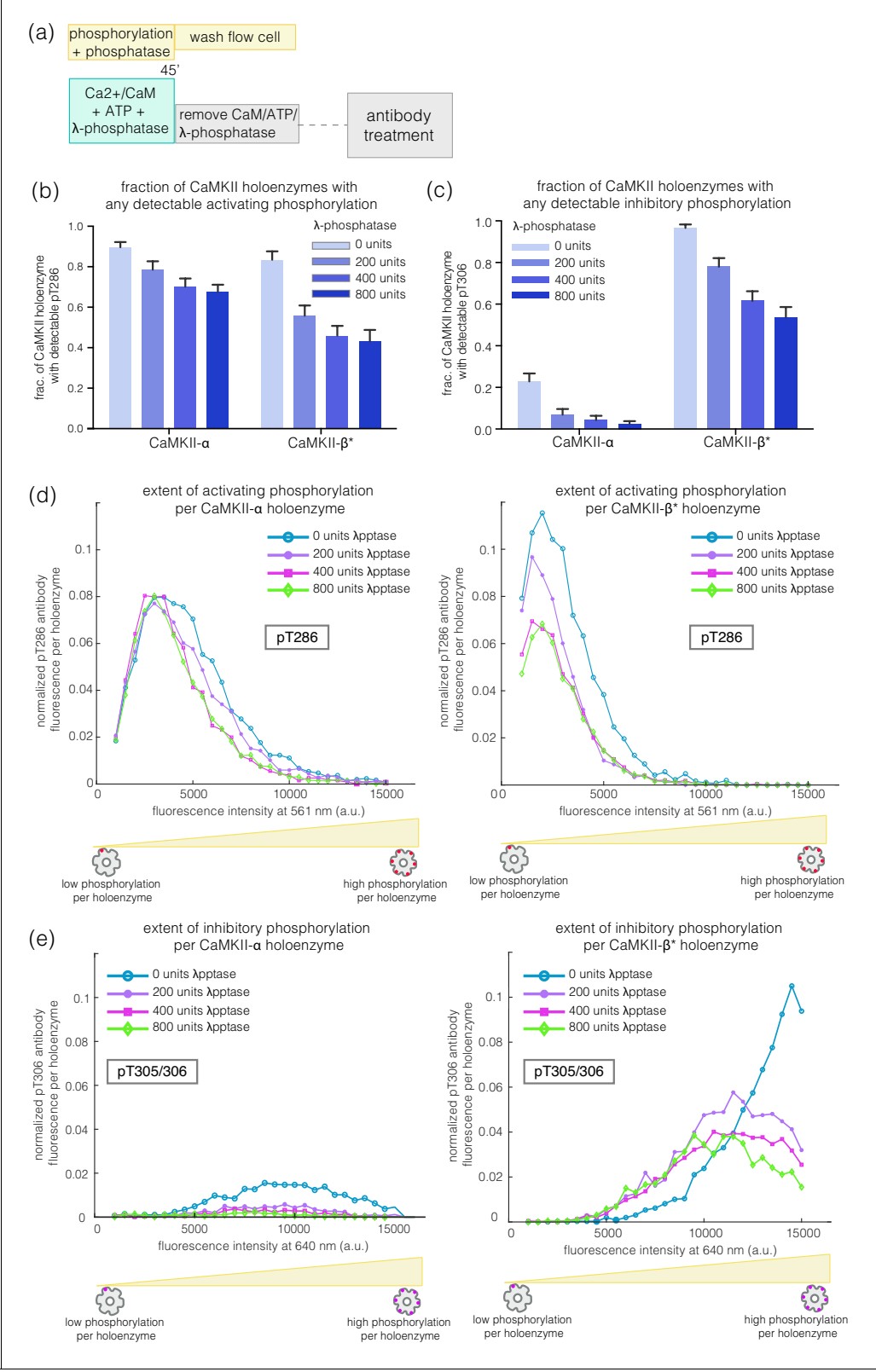

**Figure 6.** Effect of λ-phosphatase on the phosphorylation status of CaMKII when the kinase is active. (a) Schematic diagram showing the experimental set up. CaMKII was activated in the presence of 0, 200, 400, or 800 units of λ-phosphatase for 45 min. (b–c) Bar graph showing the fraction of CaMKII-α and CaMKII-β* holoenzymes that shows detectable phosphorylation at the activating site (Thr 286) and the inhibitory site (Thr 305/306), respectively, in the presence of λ-phosphatase. (d) Intensity distribution of pThr 286 (561 nm) signal for CaMKII-α (left panel) and CaMKII-β* (right panel) holoenzymes

*Figure 6 continued on next page*

*Figure 6 continued*

with detectable phosphorylation in the presence of λ-phosphatase. (e) Intensity distribution of pThr 305/306 (640 nm) signal for CaMKII-α (left panel) and CaMKII-β* (right panel) holoenzymes with detectable phosphorylation in the presence of λ-phosphatase (see Materials and methods for details of normalization).

for both isoforms (*Figure 7c–d*, *Figure 7—figure supplement 1a*). We also measured the dephosphorylation kinetics of CaMKII-α and CaMKII-β* using the catalytic subunit of protein phosphatase 1α (PP1α), an endogenous phosphatase for CaMKII. Slower rates of dephosphorylation at pThr 286 and rapid dephosphorylation at pThr 305/306 are observed for CaMKII-α and CaMKII-β* using PP1α, as for λ-phosphatase (*Figure 7—figure supplement 1b–c*).

To rule out that the resistance of pThr 286 towards rapid dephosphorylation is a consequence of on-glass treatments, we performed experiments in solution using diluted HEK 293T cell lysate (see Materials and methods) (*Figure 7—figure supplement 2a*). CaMKII-α and CaMKII-β* were first activated in the diluted cell lysate and then treated with a high concentration of staurosporine (100 μM), which inhibits kinase activity (*Figure 7—figure supplement 3*). We then inactivated calmodulin by chelating $Ca^{2+}$ by EGTA (1 mM), followed by the addition of saturating amounts of λ-phosphatase. Samples were pulled down from this reaction mixture at defined time points, and the phosphorylation at Thr 286 and Thr 305/306 was measured using the single-molecule assay. These experiments also showed slower rates of dephosphorylation at Thr 286 when compared to Thr 305/306 (*Figure 7—figure supplement 2b–c*).

The observed resistance of pThr 286 towards rapid dephosphorylation suggests that this site may be less accessible to phosphatases compared to pThr 305/306. Unexpectedly, we found that the addition of $Ca^{2+}$/CaM to phosphorylated CaMKII enhances the rate of dephosphorylation at Thr 286 by about 4-fold for both CaMKII-α and CaMKII-β* (*Figure 8a–c*, *Figure 8—figure supplement 1*). When $Ca^{2+}$/CaM is present, the pThr 286 signal is reduced at a rate that is comparable to the dephosphorylation rate of pThr 305/306. This effect does not require the addition of ATP, that is it is independent of kinase activity. The slower rates of dephosphorylation at Thr 286 is also noted for CaMKII-β, with the addition of $Ca^{2+}$/CaM reversing the protection (data not shown).

The addition of $Ca^{2+}$/CaM has no effect on the rate of dephosphorylation of pThr 305/306, indicating that $Ca^{2+}$/CaM does not increase the activity of the phosphatase. These data suggest that, in the absence of $Ca^{2+}$/CaM, the calmodulin-binding element of CaMKII plays a role in sequestering pThr 286 from phosphatases. The mechanism underlying this protection remains to be understood.

## Slower dephosphorylation at the activating site primes CaMKII for activation by subthreshold concentrations of $Ca^{2+}$/CaM

The slower rates of dephosphorylation at Thr 286 might ensure that at least some subunits in a CaMKII holoenzyme retain $Ca^{2+}$/CaM-independent activity after a calcium pulse has subsided. CaMKII subunits that are phosphorylated on Thr 286 have increased affinity for $Ca^{2+}$/CaM, which can allow a subthreshold $Ca^{2+}$-pulse to cooperatively spread Thr 286 phosphorylation within the primed holoenzyme (*De Koninck and Schulman, 1998*; *Meyer et al., 1992*; *Tse et al., 2007*). We tested this by generating a population of CaMKII holoenzymes in which only a small number of subunits are phosphorylated at Thr 286, to determine if they are primed for a response to $Ca^{2+}$/CaM.

Treatment of activated CaMKII-α with λ-phosphatase for 3–5 min generated a population in which about 50% of the holoenzymes retained some phosphorylation at Thr 286, while the rest of the holoenzymes had no detectable phosphorylation remaining (*Figure 9a–c*). The holoenzymes that do retain Thr 286 phosphorylation have reduced levels of phosphorylation per holoenzyme (*Figure 9d*). When such a sample is incubated with ATP and a sub-saturating level of $Ca^{2+}$/CaM (25 nM), the extent of phosphorylation for holoenzymes that show detectable pThr 286 is restored to the levels seen for fully activated CaMKII-α (*Figure 9a–b,d*). Treatment with ATP and sub-saturating $Ca^{2+}$/CaM does not increase the fraction of CaMKII-α holoenzymes with detectable phosphorylation at Thr 286 very much, suggesting that some extent of pre-existing phosphorylation on Thr 286 is important for gaining additional phosphorylation with sub-saturating levels of $Ca^{2+}$/CaM (*Figure 9c*).

A similar experiment using CaMKII-β* yields very different results. Here, phosphatase treatment results in substantial reduction in pThr 286 levels. Subsequent treatment with 25 nM $Ca^{2+}$/CaM and

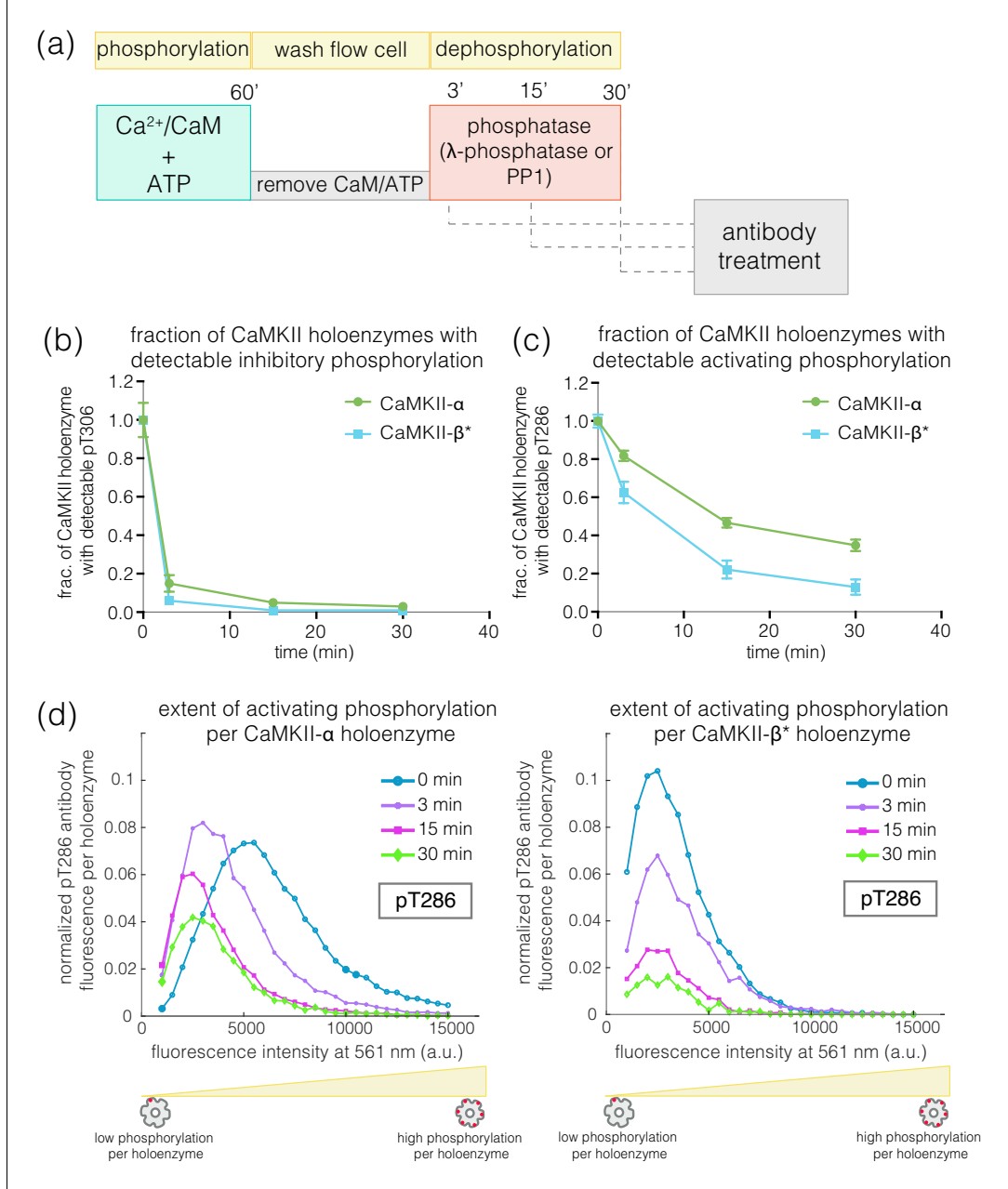

**Figure 7.** Effect of λ-phosphatase on the dephosphorylation kinetics when kinase activity is switched off. (a) Schematic diagram showing the experimental set up. CaMKII was activated, followed by a wash to remove the components of the activation buffer, and then saturating amounts of λ-phosphatase/PP1α (400–800 units) were added for 0, 3, 15, or 30 min. (b) Plot showing the fraction of α and β* holoenzymes that exhibits detectable phosphorylation at the inhibitory site (Thr 305/306) upon treatment with λ-phosphatase for defined time-points. The fractions at 3, 15, or 30 min for α and β* are normalized by the corresponding activated version that has not been exposed to any λ-phosphatase (0 min time-point, whose value is set to 1.0). (c) Same as (b) but for the activating site (Thr 286). (d) Intensity distribution for pThr 286 (561 nm) for CaMKII-α (left panel) and CaMKII-β* (right panel) holoenzymes with detectable phosphorylation after 0, 3, 15, or 30 min of λ-phosphatase treatment (see Materials and methods for details of normalization).

The online version of this article includes the following figure supplement(s) for figure 7:

**Figure supplement 1.** Effect of phosphatases on dephosphorylation kinetics when kinase activity is switched off.

**Figure supplement 2.** Measurement of dephosphorylation kinetics in solution.

**Figure supplement 3.** Inhibition of CaMKII kinase activity by staurosporine.

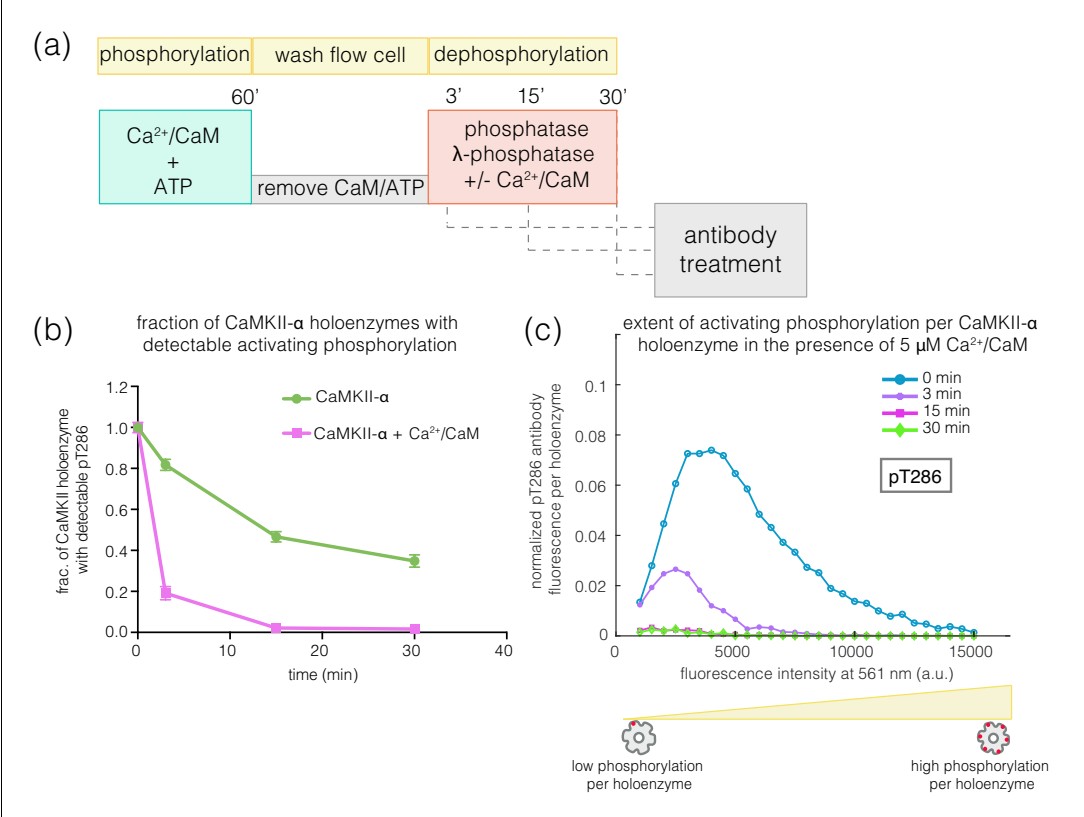

**Figure 8.** Effect of addition of Ca$^{2+}$/CaM on the rates of dephosphorylation at the activating site. (**a**) Schematic diagram showing the experimental set up. CaMKII was activated, followed by wash to remove the components of the activation buffer and then saturating amounts of λ-phosphatase were added for 3, 15, or 30 min, in the presence and absence of Ca$^{2+}$/CaM. (**b**) Plot showing the fraction of CaMKII-α holoenzymes with detectable phosphorylation at the activating site (Thr 286, right panel) after 0 min (activated control) and 3, 15, or 30 min of treatment with saturating amounts of λ-phosphatase in the absence (green trace) and presence (pink trace) of Ca$^{2+}$/CaM. The fractions at 3, 15, or 30 min are normalized with respect to activated CaMKII that has not been exposed to any λ-phosphatase (0 min, whose value is set to 1.0). (**c**) Intensity distribution for pThr 286 (561 nm) for CaMKII-α holoenzymes with detectable phosphorylation, upon 0, 3, 15, or 30 min of λ-phosphatase treatment in the presence of 5 μM Ca$^{2+}$/CaM (see Materials and methods for details of normalization).

The online version of this article includes the following figure supplement(s) for figure 8:

**Figure supplement 1.** Effect of addition of Ca$^{2+}$/CaM on the rates of dephosphorylation at the autonomy site.

ATP does not result in a marked increase in the extent of Thr 286 phosphorylation per holoenzyme. In contrast, ~40% of CaMKII-β* holoenzymes show detectable phosphorylation at Thr 305/306 after treatment with sub-saturating Ca$^{2+}$/CaM, compared to ~2% before (data not shown). Thus, in contrast to CaMKII-α holoenzymes, CaMKII-β holoenzymes tend to become inactivated when subjected to subthreshold levels of Ca$^{2+}$/CaM, rather than undergoing priming for further activation.

## Concluding remarks

CaMKII is unusual among protein kinases because it is assembled into a large holoenzyme with twelve to fourteen subunits. Analysis of the phosphorylation status of the enzyme using bulk assays, such as Western blots, has been challenging due to difficulties in purifying the holoenzymes, particularly for isoforms with longer kinase-hub linkers. In addition, bulk measurements do not provide a window into the extent of phosphorylation per holoenzyme. To overcome these limitations, we designed a single-molecule TIRF microscopy assay that relies on the capture from mammalian cell lysates of CaMKII holoenzymes tagged with fluorescent proteins. Immobilization of the captured holoenzymes in a flow-cell apparatus allows activation and dephosphorylation of CaMKII to be carried out readily, followed by measurement of the phosphorylation status using fluorescently labeled site-specific antibodies. Using this assay, we discovered that CaMKII-α and CaMKII-β, the two major

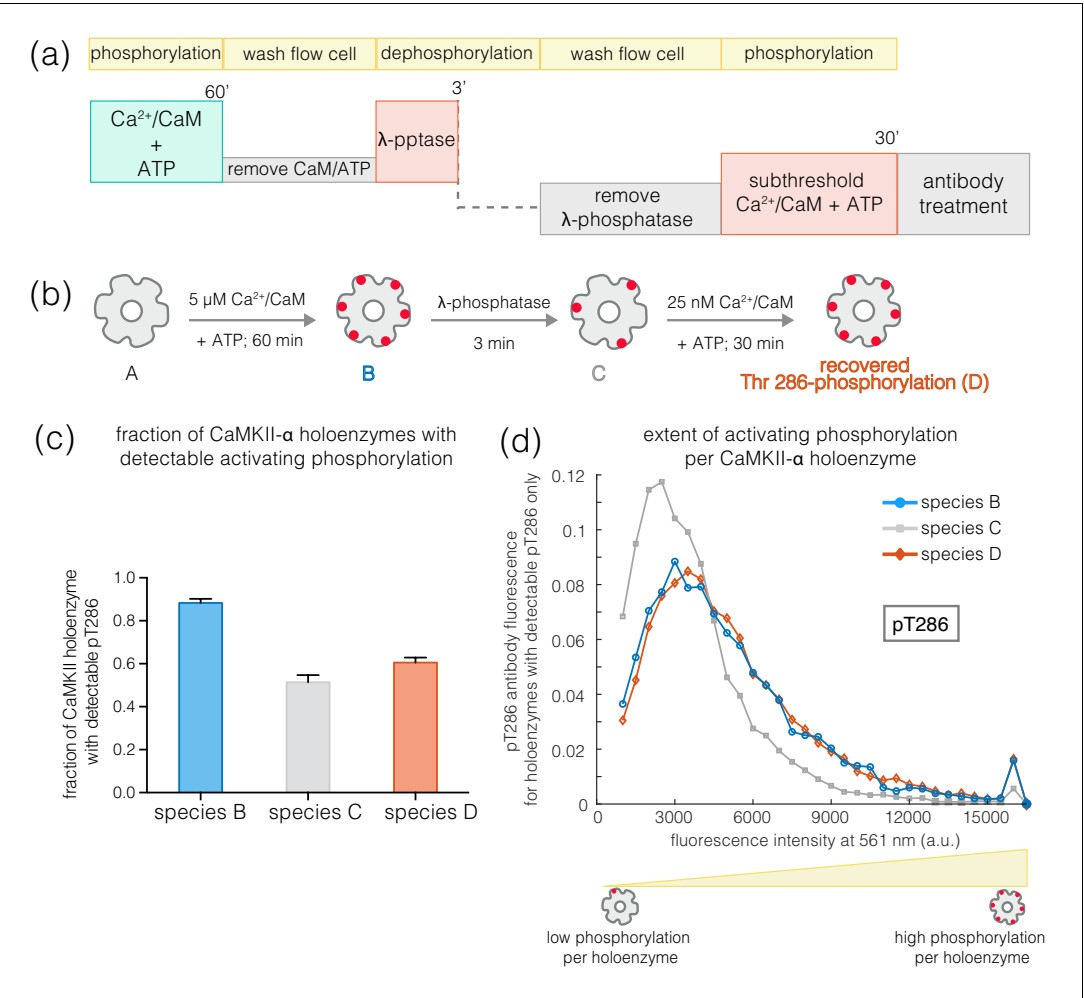

**Figure 9.** Recovery of phosphorylation at the activating site by subthreshold concentrations of $Ca^{2+}$/CaM. (**a**) Schematic diagram showing the experimental set up. CaMKII was activated, followed by a wash to remove the components of the activation buffer and then saturating amounts of λ-phosphatase were added for 3–5 min. The sample was then washed to remove the λ-phosphatase, followed by further treatment with subthreshold concentrations of $Ca^{2+}$/CaM (25 nM) for 30 min and the autophosphorylation status was measured. (**b**) Cartoon representation of the different species generated after each treatment. Each species is color-coded and the color schemes are maintained throughout the plots. (**c**) Fraction of CaMKII-α holoenzymes with detectable phosphorylation at Thr 286 is plotted for each species. (**d**) Intensity distribution of pThr 286 (561 nm) for only those CaMKII-α holoenzymes that show any detectable Thr 286 phosphorylation, for the different species of interest as described in (**b**).

isoforms in the brain, differ in the balance of activating and inhibitory phosphorylation. CaMKII-α, with a shorter linker, readily acquires activating phosphorylation, while CaMKII-β, with a longer linker, is biased towards inhibitory phosphorylation. We also tested the effects on autophosphorylation of the kinase-hub linkers from CaMKII-γ (110 residues) and CaMKII-δ (31 residues). The shorter CaMKII-δ linker behaves like that of CaMKII-α in terms of autophosphorylation outcome. The outcome for the longer CaMKII-γ linker is closer to that seen with the CaMKII-β linker.

The difference in autophosphorylation balance between CaMKII-α and CaMKII-β can be explained by the fact that Thr 286 can only be autophosphorylated in trans, between two kinase domains, since this residue is located too far from the active site of the subunit of which it is a part. Phosphorylation

of Thr 286 is known to occur predominantly through *trans*-autophosphorylation within a holoenzyme, rather than phosphorylation between different holoenzymes (*Rich and Schulman, 1998*). In contrast, inhibitory phosphorylation on Thr 305/306 can occur either in cis or in trans. We expect that the longer linker length in CaMKII-β results in a reduction in the rate of *trans*-autophosphorylation of Thr 286, while the rate of *cis*-autophosphorylation of Thr 305/306 is the same for CaMKII-α and CaMKII-β. As the kinase-hub linker length increases, the rate of activating *trans*-phosphorylation decreases, which switches the balance towards inhibitory phosphorylation in CaMKII-β. The switch is expected to be sharpened by the fact that inhibitory autophosphorylation blocks $Ca^{2+}$/CaM binding, thereby preventing activating phosphorylation.

There is a parallel between the architecture of CaMKII and A-kinase anchoring proteins (AKAPs), which are connected to cAMP-dependent protein kinase (PKA) by long flexible linkers. It has been suggested that the length and flexibility of the AKAP linker determines the spatial range over which PKA can phosphorylate its substrates (*Smith et al., 2013*). Since CaMKII holoenzymes are anchored within the post-synaptic density, the differences in the lengths of the kinase-hub linkers in different isoforms may likewise control the spatial extent of substrate phosphorylation enabled by the activation of a particular CaMKII holoenzyme. Our work has now revealed a key difference between CaMKII and AKAP:PKA complexes, which is that the kinase-hub linker length in CaMKII can control the activation status of the holoenzyme, not just the spatial range of the kinase.

Our experiments show that activating phosphorylation on Thr 286 is resistant to dephosphorylation, whereas inhibitory phosphorylation on Thr 305/306 is reversed rapidly. This suggests that the phosphate group on Thr 286 is somehow protected from phosphatases. The protection afforded to pThr 286 is released by the addition of $Ca^{2+}$/CaM, suggesting that the calmodulin-binding element of CaMKII adopts a conformation that protects pThr 286 in the absence of $Ca^{2+}$/CaM. The presence of such a conformational change may explain the apparent discrepancy between our results and studies reporting that pThr 286 undergoes very rapid dephosphorylation in neurons (*Lee et al., 2009*). The conclusion that pThr 286 is rapidly dephosphorylated relied on the utilization of a FRET-based sensor, Camui (*Takao et al., 2005*), which reports on conformational changes undergone by CaMKII, and does not report on phosphorylation directly. It will be interesting to see whether the release of $Ca^{2+}$/CaM from CaMKII after activation results in a compaction of CaMKII that might explain both the protection of pThr 286 that we observe and the rapid relaxation back to a compact state that is reported by the Camui sensor.

The observed resistance of pThr 286 to rapid dephosphorylation by phosphatases can enable CaMKII activity to persist for some time after the decay of $Ca^{2+}$ pulses. In addition, CaMKII subunits that are phosphorylated on Thr 286 have higher affinity for $Ca^{2+}$/CaM, which can facilitate the rebinding of $Ca^{2+}$/CaM during subsequent pulses of calcium. Due to the cooperativity of $Ca^{2+}$/CaM binding to CaMKII, such a priming mechanism enables CaMKII holoenzymes that have retained some activating phosphorylation to rapidly reacquire higher levels of phosphorylation, even in the presence of subthreshold concentrations of $Ca^{2+}$/CaM. This may enable the history of synaptic activity, including the frequency and the gaps between periods of activity, to be encoded in CaMKII.

There are differences in the expression levels of CaMKII-α and CaMKII-β in neurons and these levels can be regulated by synaptic activity (*Thiagarajan et al., 2002*). Differences in these levels can result in differences in cellular targeting and stimulus frequency responses (*De Koninck and Schulman, 1998*; *Shen et al., 1998*). We now show that expression level differences between isoforms can also result in changing the balance of activating and inhibitory autophosphorylation. The lower inhibitory autophosphorylation of CaMKII-α relative to that in CaMKII-β can have a number of consequences. Autophosphorylation of the inhibitory site reduces binding of CaMKII-α to the postsynaptic density (PSD) in vitro (*Strack et al., 1997*). By contrast, abrogation of such autophosphorylation greatly reduces dissociation of the kinase from PSD in neurons (*Shen and Meyer, 1999*) and increases PSD binding in transgenic animals, thereby altering synaptic plasticity and long-term potentiation (*Elgersma et al., 2002*). The rapid inhibitory phosphorylation of CaMKII-β, on the other hand, limits binding of $Ca^{2+}$/CaM, which is required for dissociation of the kinase from F-actin (*Shen and Meyer, 1999*; *Wang et al., 2019*), an important step in structural remodeling of the synapse (*Kim et al., 2015*). Our results show that by varying the relative expression levels of α and β-isoforms, the cell can generate CaMKII heterooligomers with variable response to activating signals, thereby fine-tuning the response of CaMKII to $Ca^{2+}$ spike trains in the brain.

# Materials and methods

## Key resources table

| Reagent type (species) or resource | Designation | Source or reference | Identifiers | Additional information |
|---|---|---|---|---|
| Gene (*human*) | CaMKII-α | | Uniprot_ID: Q9UQM7 | |
| Gene (*human*) | CaMKII-β | | Uniprot_ID: Q13554 | |
| Gene (*human*) | CaMKII-β′E | | Uniprot_ID: Q13554-3 | |
| Gene (*human*) | CaMKII-γ | | Uniprot_ID: Q13555 | |
| Gene (*human*) | CaMKII-δ | | Uniprot_ID: Q13557 | |
| Recombinant DNA reagent | pET21a-BirA | Addgene | plasmid # 20857 | for biotinylation of the AVI tag in HEK 293T cells |
| Recombinant DNA reagent | EYFP-CaM | Addgene | plasmid # 47603 | coexpressed for in cell activation of CaMKII |
| Recombinant DNA reagent | pEGFP-C1 (plasmid) | Clontech, Mountain View, CA | | vector backbone for inserting the CaMKII genes |
| Recombinant DNA reagent | pSNAP$_f$ (plasmid) | New England Biolabs, MA | N9183S | vector backbone |
| Cell line (*human*) | HEK 293T | UC Berkeley cell culture facility | | authenticated using STR profiling and tested negative for mycoplasma |
| Antibody | anti-CaMKII (phospho T286); mouse monoclonal | Abcam | ab171095 | 1:500 |
| Antibody | anti-CaMKII (phospho T306); rabbit polyclonal | PhosphoSolutions | p1005-306 | 1:500 |
| Peptide, recombinant protein | Poly-L-lysine PEG (PLL:PEG) | SuSoS, Dübendorf, Switzerland | PLL(20)-g[3.5]- PEG(2) | preparation of flow chambers |
| Peptide, recombinant protein | streptavidin | Sigma-Aldrich | S0677 | functionalize the glass substrates for capturing biotinylated CaMKII |
| Peptide, recombinant protein | calmodulin | Sigma-Aldrich | C4874 | activation of CaMKII |
| Peptide, recombinant protein | λ-phosphatase | New England Biolabs, MA | P0753L | phosphatase |
| Peptide, recombinant protein | PP1α | EMD Millipore, Burlington, MA | 14–595 | phosphatase |
| Chemical compound, drug | PEG-Biotin | SuSoS, Dübendorf, Switzerland | PLL(20)-g[3.5]-PEG(2)/PEG(3.4)-biotin(50%) | preparation of flow chambers |
| Chemical compound, drug | 1% protease inhibitor cocktail | Sigma | P8340 | protease inhibitor cocktail for lysis buffer |
| Chemical compound, drug | 0.5% phosphatase inhibitor cocktail 2 and 3 | Sigma | P0044 and P5726 | phosphatase inhibitor cocktails for lysis buffer |
| Chemical compound, drug | staurosporine | Abcam | ab120056 | kinase inhibitor |
| Chemical compound, drug | cyclosporin A | Sigma-Aldrich | 30024 | phosphatase inhibitor |
| Chemical compound, drug | okadaic acid | Abcam | ab141831 | phosphatase inhibitor |

*Continued on next page*

*Continued*

| Reagent type (species) or resource | Designation | Source or reference | Identifiers | Additional information |
|---|---|---|---|---|
| Software, algorithm | FIJI (ImageJ) | open access software, see https://imagej.net/Fiji/Downloads | | image processing |
| Software, algorithm | in-house Matlab codes | open access, see *Source code 1* | | image processing |
| Other | sticky-Slide VI 0.4 | Ibidi | 80608 | flow chambers |
| Other | glass coverslips | Ibidi | 10812 | functionalized substrates |

## Preparation of plasmids

Human CaMKII-α (Uniprot_ID: Q9UQM7) and human CaMKII-β (Uniprot_ID: Q13554) were cloned into the pEGFP-C1 vector backbone (Clontech, Mountain View, CA), after modifying the vector to contain a biotinylation sequence (Avitag, GLNDIFEAQKIEWHE) followed by a linker (GASGASGAS-GAS) at the N-terminus of mEGFP. CaMKII-α or CaMKII-β was cloned at the C-terminus of mEGFP, with a linker sequence (PreScission protease site: LEVLFQGP) separating the mEGFP tag from the coding sequence of CaMKII-α/CaMKII-β (i.e., the final construct is organized as Avitag-linker-mEGFP-linker-CaMKII-α/CaMKII-β). These constructs were then used as a template to produce the other CaMKII-variants. The splice-variant of CaMKII-β (CaMKII-β'E) was produced by deleting 155 residues the β-isoform (Uniprot_ID: Q13554-3). CaMKII-β*, CaMKII-γ*, CaMKII-δ*, and CaMKII-β'E* were produced by replacing the linker in CaMKII-α with that from CaMKII-β (Uniprot_ID: Q13554), CaMKII-γ (Uniprot_ID: Q13555), CaMKII-δ (Uniprot_ID: Q13557), and CaMKII-β'E, respectively (*Figure 1—figure supplement 1*). mCherry-tagged variants were generated by replacing the mEGFP tag with mCherry in these constructs. pET21a-BirA was a gift from Alice Ting (Addgene plasmid # 20857). BirA was cloned into the pSNAP$_f$ vector (New England Biolabs, MA) after modifying the vector backbone to remove the SNAP-tag. EYFP-CaM was a gift from Emanuel Strehler (Addgene plasmid # 47603). The EYFP tag was removed from this vector to generate the mammalian expression construct for calmodulin. All constructs with large domain insertions and deletions were made using standard protocols for Gibson assembly (New England Biolabs, MA). All point mutants used were generated using the standard Quikchange protocols (Agilent Technologies, Santa Clara, CA).

## Tissue culture and DNA transfection

HEK 293T cells were obtained from the UC Berkeley cell culture facility (authenticated using STR profiling and tested negative for mycoplasma). These cells were grown in Dulbecco's Modified Eagle Medium + GlutaMaX (DMEM, Gibco, Thermo Fisher) that is supplemented with 10% FBS (Avantor Seradigm, VWR, Radnor, Pennsylvania), Antibiotic-Antimycotic (AA, Thermo Fisher) at 100X dilution and 20 mM HEPES buffer and maintained at 37°C under 5% $CO_2$. Transient transfection of CaMKII variants were done using the standard calcium phosphate protocol (*Wigler et al., 1977*). Briefly, CaMKII plasmids (200 ng/400 ng/800 ng depending on the construct) were mixed with 6 μg of empty pcDNA3.1 vector and 1 μg of BirA. This DNA mix was then diluted with ddH$_2$O (10X), 250 mM $CaCl_2$ was added and the mixture was allowed to sit for 15 min at room temperature. Following this incubation, 2X HBS buffer (50 mM HEPES, 280 mM NaCl, 1.5 mM $Na_2HPO_4$, pH 7.1) was added to it dropwise and mixed thoroughly by reverse pipetting. This mixture was then added to the HEK 293T cells and the cells were allowed to express the protein for 18–20 hours before harvesting.

## Preparation of flow cells for single-molecule microscopy

All single-molecule experiments were performed in flow chambers (sticky-Slide VI 0.4, Ibidi, Planegg, Germany) that were assembled with functionalized glass substrates (Ibidi glass coverslips, bottom thickness 170 μm+/–5 μm). The glass substrates were first cleaned using 2% Hellmanex III solution (Hellma Analytics) for 30 min, followed by a 30 min sonication in 1:1 mixture (vol/vol) of isopropanol: water. The glass substrates were then dried with nitrogen and cleaned for another 5 min in a plasma cleaner (Harrick Plasma PDC-32 G, Ithaca, NY). These cleaned glass substrates were used to

assemble the flow chambers immediately after plasma cleaning. After assembly, the glass substrates were treated with a mixture of Poly-L-lysine PEG and PEG-Biotin (1000:1, both at 1 mg/mL) for 30 min (SuSoS, Dübendorf, Switzerland). The glass substrates were then washed with 2 mL of Dulbecco's phosphate-buffered saline (DPBS, Gibco, Thermo Fisher). Streptavidin (Sigma-Aldrich, S0677) was added to these glass substrates at a final concentration of 0.1 mg/mL and incubated for 30 min. Following incubation, excess streptavidin was washed away using 2 mL of DPBS and these assembled flow chambers with streptavidin-coated glass substrates were used for all our single-molecule experiments.

## Cell lysis and pulldown of biotinylated CaMKII in flow chambers

CaMKII variants were allowed to express for 18–20 hours before they were harvested. The co-expression of the *E. coli* biotin ligase, BirA, with the CaMKII variants bearing an Avitag, results in the biotinylation of CaMKII in HEK 293T cells. After harvesting, the cells were lysed in a lysis buffer containing 25 mM Tris at pH 8, 150 mM KCl, 1.5 mM TCEP-HCl, 1% protease inhibitor cocktail (P8340, Sigma), 0.5% phosphatase inhibitor cocktail 2 (P0044, Sigma) and 3 (P5726, Sigma), 50 mM NaF, 15 μg/ml benzamidine, 0.1 mM phenylmethanesulfonyl fluoride and 1% NP-40 (Thermo Fisher). The cell lysate was then diluted 100–200 times in DPBS and 100 μL of this diluted cell lysate was added to a well in the flow chamber for 1 min, before washing it out with 1 mL of DPBS. During this incubation, the biotinylated mEGFP-CaMKII variants were immobilized on the surface of the functionalized glass substrates, owing to the streptavidin-biotin interaction. A buffer exchange was then done in the flow chamber to replace the DPBS with the gel filtration buffer (25 mM Tris, 150 mM KCl, 1.5 mM TCEP, pH 8).

## Activation of CaMKII on glass substrates

The glass-immobilized CaMKII holoenzymes were activated by flowing in an activation buffer (with a final concentration of 100 μM $CaCl_2$, 10 mM $MgCl_2$, 500 μM ATP in the gel filtration buffer and the CaM concentration varies between 0.02–5 μM CaM (Sigma-Aldrich, C4874) depending on the experiment) into the flow chambers for 60 min. Following this incubation, the activation buffer was washed out using 2 mL of the gel filtration buffer. All experiments have been replicated at least 3–5 times using samples prepared over different days.

## Activation of CaMKII in HEK 293T cells

HEK 293T cells overexpressing CaMKII and CaM were washed with HBSS buffer (Gibco, Thermo Fisher, 14170112) plus 25 mM HEPES and incubated at room temperature for 5–10 min. The cells were then treated with 10 μM ionomycin (Sigma-Aldrich, 407953) and 3 mM $CaCl_2$ (final concentration) for 10–15 min, activating CaMKII within the cell. Unactivated samples were obtained from cells treated with HBSS buffer alone for the same amount of time. The cells were then lysed, and activated or unactivated CaMKII was pulled down on functionalized glass substrates without further treatment. For experiments that involved incubation with phosphatase inhibitors, 250 nM cyclosporin A (Sigma-Aldrich, 30024) and 250 nM okadaic acid (Abcam, ab141831) were added to the cells for 60 min prior to the activation of CaMKII. The same procedure as above was then followed to activate CaMKII in cells and pull down activated or unactivated CaMKII directly from the cell lysates.

## Immunofluorescence assay with phosphospecific antibodies

The autophosphorylation status of the CaMKII holoenzyme at the activating (Thr 286) and the inhibitory (Thr 305/306) sites was estimated using phosphospecific antibodies for pThr 286 (Abcam: ab171095) and pThr 305/306 (PhosphoSolutions: p1005-306). The immobilized, activated and unactivated CaMKII holoenzyme was incubated with a mixture of these two phosphospecific primary antibodies (in a 1:1 ratio using a 1:500 dilution in 5% (w/v) BSA) for 45 min. Subsequently, excess primary antibodies were washed out with 3 mL DPBS. This was followed by a 30 min incubation with a 1:1 mix of Alexa-labeled secondary antibodies (1:1000 dilution in 5% BSA), that are complementary to the primary antibodies used. Anti-mouse secondary antibody labeled with Alexa-594 (Cell Signaling Technology) was used for pThr 286-specific primary antibody; anti-rabbit secondary antibody labeled with Alexa-647 (Cell Signaling Technology, Danvers, MA) was used for the pThr 305/306-specific primary antibody. After incubation with the secondary antibody, the flow chambers were

washed again using 3 mL of DPBS and the samples were imaged using Total Internal Reflection Fluorescence (TIRF) microscopy.

## Validation of the primary antibodies

The two primary antibodies for pThr 286 and pThr 305/306 do not cross-react with each other's target sites, as shown by mutations in their respective epitopes ($T^{286}$ is mutated to $A^{286}$ and the epitope for the Thr 305/306 primary antibody is changed from $T^{305}T^{306}$MLATRNFS to $A^{305}V^{306}$I) in CaMKII-α (*Figure 2—figure supplement 1*). These primary antibodies also do not show any reactivity with the unactivated forms of CaMKII-α or CaMKII-β, as pulled down from the cell lysate and before any treatment with activation buffer (data not shown). Additionally, the primary antibody incubation time was optimized so that the phosphosites were nearly saturated by their respective antibodies and we chose the incubation time to be 45 min in all our assays (data not shown).

## Phosphatase assay when the kinase is active

To test how the presence of phosphatase affects the phosphorylation status of CaMKII when the kinase is active, increasing amounts of λ-phosphatase (200–800 units, New England Biolabs) and 1 mM MnCl$_2$ was added to samples of glass-immobilized CaMKII, in the presence of activation buffer containing 5 μM Ca$^{2+}$/CaM. 1 unit of λ-phosphatase is defined as the amount of enzyme that hydrolyzes 1 nmol of p-nitrophenyl phosphate in 1 min at 30°C (New England Biolabs). After 45 min of incubation, the flow chambers were washed with 2 mL DPBS to remove the λ-phosphatase and the activation buffer. The phosphorylation of Thr 286 and Thr 305/306 was then examined using the immunofluorescence assay described above. No further reduction in phosphorylation was observed upon adding more than 400 units of λ-phosphatase and this is considered as a saturating amount of phosphatase.

## Phosphatase assay when the kinase activity is switched off

To test the sensitivity of the two autophosphorylation sites to phosphatases in the absence of kinase activity, activated CaMKII was treated with two different phosphatases, λ-phosphatase (New England Biolabs) and PP1α (EMD Millipore, Burlington, MA). A phosphatase buffer that contains 1 mM MnCl$_2$ and 400–800 units of λ-phosphatase or 800 units of PP1α was added to the flow chambers displaying activated CaMKII, after the activation buffer had been washed out. 1 unit of PP1α is defined as the amount of enzyme that releases 1 nmol phosphate per minute from the phosphorylated substrate DiFMUP (6,8-difluoro-4-methylumbelliferyl phosphate) (EMD Millipore). The dephosphorylation reactions were carried out for defined periods of time (3, 15, or 30 min), following which the phosphatase buffer was washed away with 2 mL DPBS and the autophosphorylation states were examined using the immunofluorescence assay described above.

## Phosphatase assay in solution

The HEK 293T cell lysate was diluted by 1:100–200. CaMKII was activated in this diluted lysate (in solution) by adding an activation buffer containing 5 μM Ca$^{2+}$/CaM and 5 mM Tris-buffered TCEP (Sigma-Aldrich). After activation for 45 min, 100 μM staurosporine (Abcam, ab120056) was added to this reaction mix to inhibit the kinase activity. After inhibiting CaMKII for 10 min, saturating amounts of λ-phosphatase (400 units) was added to the solution. Samples were pulled down after 3 or 15 min of incubation and the phosphorylation status on Thr 286 and Thr 305/306 was measured. We verified that 100 μM staurosporine was efficient in switching off kinase activity by activating CaMKII-α/β* in the presence of the inhibitor. This treatment completely abolished any autophosphorylation of Thr 286 or Thr 305/306 (*Figure 7—figure supplement 3*).

## Thr 286-phosphorylation recovery assay for CaMKII priming

Activated samples of glass-immobilized CaMKII were treated with saturating amounts of λ-phosphatase and 1 mM MnCl$_2$ for 3–5 min, followed by washing away of the phosphatase buffer with 2 mL of DPBS. This sample was then incubated with an activation buffer (see details above) containing subthreshold concentrations of 25 nM Ca$^{2+}$/CaM for 30 min. The activation buffer was aspirated out with 2 mL DPBS and the phosphorylation status at the activating and inhibitory sites were measured.

## Single-molecule total internal reflection fluorescence (TIRF) microscopy

Single-particle total internal reflection fluorescence images were acquired on a Nikon Eclipse Ti-inverted microscope equipped with a Nikon 100 × 1.49 numerical aperture oil-immersion TIRF objective, a TIRF illuminator, a Perfect Focus system, and a motorized stage. Images were recorded using an Andor iXon electron-multiplying charge-coupled device camera. The sample was illuminated using the LU-N4 laser unit (Nikon, Tokyo, Japan) with solid state lasers for the 488 nm, 561 nm and 640 nm channels. Lasers were controlled using a built-in acousto-optic tunable filter (AOTF). The 405/488/561/638 nm Quad TIRF filter set (Chroma Technology Corp., Rockingham, Vermont) was used along with supplementary emission filters of 525/50 m, 600/50 m, 700/75 m for 488 nm, 561 nm, 640 nm channel, respectively. 3-color image acquisition was performed by computer-controlled change of illumination and filter sets at 42 different positions from an initial reference frame, so as to capture multiple non-overlapping images. Image acquisition was done using the Nikon NIS-Elements software. mEGFP-CaMKII, Alexa-594-labeled anti-Thr 286 antibody, and Alexa-647-labeled anti-Thr 305/306 antibody were imaged by illuminating 488 nm laser set to 5.2 mW, 561 nm laser set to 6.9 mW, and 640 nm laser set to 7.8 mW, respectively. The laser power was measured with the field aperture fully opened. Images were acquired using an exposure time of 80 milliseconds for 488 nm and 561 nm, and an exposure time of 100 milliseconds for 640 nm. The only exception was for acquiring mCherry single-molecule images, where an exposure of 200 milliseconds was used. Epifluorescence images were also acquired using a Nikon Eclipse Ti-inverted microscope with a Nikon 20x objective and an Andor iXon electron-multiplying charge-coupled device camera. A mercury arc lamp (Lumencor Tech., Beaverton, OR) was used for epifluorescence illumination at 488 and 561 nm.

## Analyses of single-molecule TIRF data

Individual single particles in all three channels were detected and localized using the single particle tracking plugin TrackMate in ImageJ (*Jaqaman et al., 2008*; *Schindelin et al., 2012*). The particles were localized with the Difference of Gaussian (DoG) detector with an initial diameter set to six pixels, typically leading to the detection of 25000–50000 particles per sample. The detection threshold value in TrackMate is set to 12 or 15 depending on the wavelength at which the image is acquired. The particles outside the center area of $350 \times 350$ pixel$^2$ were excluded due to heterogeneous TIRF illumination. No further filtering processes was applied in TrackMate.

Intensity distribution and fraction of holoenzymes showing any detectable phosphorylation were analyzed using custom in-house software written in MATLAB (these programs are provided as a source code file). For analyzing the fraction of holoenzymes that show any detectable phosphorylation at a given phosphosite, the XY-coordinates for each particle were extracted from the TrackMate data. For each mEGFP-CaMKII, distances to all the antibody particles were computed, and colocalization was identified if the inter-particle distance was below two pixels. The fraction of CaMKII holoenzymes (i.e., mEGFP-CaMKII with 488-channel intensity greater than 2500 arbitrary units) with detectable phosphorylation was computed as a ratio of the number of CaMKII colocalized with antibody to the total number of CaMKII holoenzymes. In a typical case, false-positive colocalization due to coincidentally neighbored particles was less than 0.5%, as determined from analyzing multiple, unrelated sets of images with comparable particle density. The intensity values acquired from TrackMate statistics data for the pThr 286 and pThr 305/306 antibody-channels were used to calculate an intensity histogram, for the colocalized (i.e., corresponding to holoenzymes that show detectable phosphorylation) and the non-colocalized spots (i.e., corresponding to holoenzymes that do not show any detectable phosphorylation with intensity values in the phosphospecific antibody channel being zero). This results in scaling of the area under the histogram for the antibody intensity by a factor that takes into account the fraction of holoenzymes that show no detectable phosphorylation. A bin width of 500 was used for computing the histogram. After calculating the distribution, we plotted the intensity histogram for only the colocalized spots and an intensity cutoff of about 15000 arbitrary units.

For analysis of samples where mEGFP-CaMKII-α and mCherry-CaMKII-β* were co-transfected (i.e., α:β* ratio equals 3:1 and 1:1), we first identified the heterooligomeric holoenzyme population (~80% of the total holoenzymes, data not shown) and then calculated what fraction of these heterooligomers showed detectable inhibitory phosphorylation. An intensity histogram was also plotted for the pThr 305/306 signal for these holoenzymes, as described above.

## Acknowledgements

We thank members of the Kuriyan lab, especially Laura Nocka for helpful discussions and Miles Tuncel for help with cloning and experimental setup. pET21a-BirA was a gift from Alice Ting (Addgene plasmid # 20857). EYFP-CaM was a gift from Emanuel Strehler (Addgene plasmid # 47603). We thank Darren McAffee for helpful discussions regarding data analysis. MB thanks NIGMS (K99 GM 126145) for funding.

## Additional information

### Competing interests

John Kuriyan: Senior editor, *eLife*. The other authors declare that no competing interests exist.

### Funding

| Funder | Grant reference number | Author |
| --- | --- | --- |
| National Institute of General Medical Sciences | K99 GM 126145 | Moitrayee Bhattacharyya |
| Howard Hughes Medical Institute | | John Kuriyan |

The funders had no role in study design, data collection and interpretation, or the decision to submit the work for publication.

### Author contributions

Moitrayee Bhattacharyya, Conceptualization, Resources, Data curation, Software, Formal analysis, Funding acquisition, Validation, Investigation, Visualization, Methodology, Writing - original draft, Writing - review and editing; Young Kwang Lee, Resources, Data curation, Software, Formal analysis, Validation, Investigation, Methodology, Writing - original draft, Writing - review and editing; Serena Muratcioglu, Data curation, Formal analysis, Investigation, Writing - original draft, Writing - review and editing; Baiyu Qiu, Priya Nyayapati, Data curation, Formal analysis, Investigation; Howard Schulman, Resources, Formal analysis, Validation, Writing - original draft, Writing - review and editing; Jay T Groves, John Kuriyan, Conceptualization, Resources, Formal analysis, Supervision, Funding acquisition, Validation, Investigation, Visualization, Methodology, Writing - original draft, Project administration, Writing - review and editing

### Author ORCIDs

Moitrayee Bhattacharyya https://orcid.org/0000-0002-2168-1541
Young Kwang Lee https://orcid.org/0000-0003-0056-6357
John Kuriyan https://orcid.org/0000-0002-4414-5477

### Decision letter and Author response

Decision letter https://doi.org/10.7554/eLife.53670.sa1
Author response https://doi.org/10.7554/eLife.53670.sa2

## Additional files

### Supplementary files

• Source code 1. In-house Matlab programs that are used for data analyses are provided as an open source package. A readme file and a test dataset is included for clarity.

• Transparent reporting form

## Data availability

All data generated or analyzed during this study are summarized in the manuscript, figures, appendix, and supplementary files. The in-house Matlab programs that are used for data analysis are provided as Source code file 1 and is open-access.

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

## Appendix 1

# A kinetic model to explain the switch between activating and inhibitory autophosphorylation tendencies between CaMKII-α and CaMKII-β

A simple kinetic model is built to explain the observed differences in the balance between activating and inhibitory autophosphorylation. The model is based on a CaMKII dimer, which avoids the extremely large number of intermediate states that need to be considered for a CaMKII dodecamer. Phosphorylation at the inhibitory site (Thr 305/306) can occur both in cis and in trans, whereas Thr 286 can only be phosphorylated in trans.

We wrote a series of chemical reactions that can lead to autophosphorylation at Thr 286 and Thr 305/306. A representative set of values for the forward and reverse rate constants are shown below. The reaction kinetics corresponding to these equations were simulated using Berkeley Madonna, a general-purpose differential equation solver (http://www.berkeleymadonna.com/). We monitored the accumulation of all species that are phosphorylated at either Thr 286 or Thr 305/306 under different rates of *trans*-phosphorylation, keeping the rates of *cis*-phosphorylation constant (***Figure 5—figure supplement 1a–b***).

The on and off rates for $Ca^{2+}$/CaM are set to $10^5 \, M^{-1} \, sec^{-1}$ (denoted onrate) and $10 \, sec^{-1}$ (denoted offrate) respectively, corresponding to a dissociation constant of $10^{-4} \, M^{-1}$ prior to CaM trapping. We mimicked the effect of CaM trapping by ignoring the dissociation of $Ca^{2+}$/CaM from species that are phosphorylated at Thr 286. We ignore the effects of phosphatases in this simulation, and set the rate of dephosphorylation to be negligible ($10^{-9} \, sec^{-1}$, denoted reverserate). The initial concentrations of CaM and CaMKII are set to $10^{-4} \, M$ and $10^{-9} \, M$, respectively. The equations considered are summarized in ***Appendix 1—figure 1***, where K denotes the kinase (CaMKII) and C denotes CaM. Phosphorylation on Thr 286 or Thr 305/306 is denoted by 286 and 306, respectively.

Reaction 1:  $KK + C \underset{\text{offrate}}{\overset{\text{onrate}}{\rightleftharpoons}} KKC$

Reaction 7:  $K306KC \underset{\text{onrate}}{\overset{\text{offrate}}{\rightleftharpoons}} K306K + C$

Reaction 2:  $KKC + C \underset{\text{offrate}}{\overset{\text{onrate}}{\rightleftharpoons}} KCKC$

Reaction 8:  $KKC286 \underset{\text{reverserate}}{\overset{\text{rate306}}{\rightleftharpoons}} K306KC286$

Reaction 3:  $KCKC \underset{\text{reverserate}}{\overset{\text{rate286}}{\rightleftharpoons}} KCKC286$

Reaction 9:  $KK \underset{\text{reverserate}}{\overset{\text{ratecis306}}{\rightleftharpoons}} K306K$

Reaction 4:  $KCKC286 \underset{\text{reverserate}}{\overset{\text{rate286}}{\rightleftharpoons}} KC286KC286$

Reaction 10:  $K306K \underset{\text{reverserate}}{\overset{\text{ratecis306}}{\rightleftharpoons}} K306K306$

Reaction 5:  $KCKC286 \underset{\text{onrate}}{\overset{\text{offrate}}{\rightleftharpoons}} KKC286 + C$

Reaction 11:  $KKC \underset{\text{reverserate}}{\overset{\text{ratecis306}}{\rightleftharpoons}} K306KC$

Reaction 6:  $KKC \underset{\text{reverserate}}{\overset{\text{rate306}}{\rightleftharpoons}} K306KC$

Reaction 12:  $KKC286 \underset{\text{reverserate}}{\overset{\text{ratecis306}}{\rightleftharpoons}} K306KC286$

**Appendix 1—figure 1.** A summary of the reactions considered in the kinetic model. K denotes the kinase (CaMKII) and C denotes CaM. Phosphorylation on Thr 286 or Thr 305/306 is denoted by 286 and 306, respectively.

We ran the kinetic simulations for two sets of parameters. For the first set, the value of $k_{cat}$ for the kinase reaction is set to $1 \, sec^{-1}$ and $0.1 \, sec^{-1}$ for the *trans*-phosphorylation of Thr 286 (denoted rate286) and Thr 305/306 (denoted rate306), respectively. The rate of *cis*-phosphorylation of Thr 305/306 (denoted ratecis306) is set to $0.1 \, sec^{-1}$. Under these

conditions, the simulation shows a dominant accumulation of species that are phosphorylated on Thr 286 (*Figure 5—figure supplement 1a*). Next, to mimic the effect of increasing the kinase-hub linker length, the rate of *trans*-phosphorylation was reduced by 10-fold for both Thr 286 and Thr 305/306 (i.e., 0.1 sec$^{-1}$ and 0.01 sec$^{-1}$, respectively), while keeping the rate of *cis*-phosphorylation unaltered. Under these conditions, the simulations show a higher accumulation of the inhibitory phosphorylation (*Figure 5—figure supplement 1b*).

