## [Decision Letter]

**Acceptance summary:**

The studies provide new insights into the mechanisms of activation and inactivation of CaMKII in response to elevated calcium. They highlight a role for the length of the linker between the catalytic domain and the hub of CaMKII in determining the extent to which an activating versus an inhibitory autophosphorylation occurs in response to calcium, and explains why the kinetics of activation and inactivation of the CaMKII-α and CaMKII-β isoforms are different. Overall, this paper provides a significant advance in our understanding of the regulation of this key protein kinase.

**Decision letter after peer review:**

Thank you for submitting your article "Flexible linkers in CaMKII control the balance between activating and inhibitory autophosphorylation" for consideration by *eLife*. Your article has been reviewed by three peer reviewers, and the evaluation has been overseen by Tony Hunter as the Reviewing Editor and Jonathan Cooper as the Senior Editor. The following individuals involved in review of your submission have agreed to reveal their identity: John D Scott (Reviewer #1); Stefan Knapp (Reviewer #2).

Our decision has been reached after consultation between the three reviewers. Based on these discussions, we are pleased to inform you that we will offer you the opportunity of submitting a revised version of your paper to *eLife*.

The new insights you have obtained through your biochemical, single molecule studies of the mechanisms of activation and inactivation of CaMKII in response to elevated calcium, in which the length of the linker between the central hub and the catalytic domain dictates whether autophosphorylation is primarily at the T286 activating site or at the T305/306 inhibitory sites, provide a significant advance in our understanding of the regulation of this key enzyme. Your studies have led to the formulation of an intricate model for the activation and inactivation of CaMKII's that may prove challenging for some readers to fully appreciate, but the in vitro data are of sufficient interesting for publication in *eLife*, and they may stimulate more research focusing on the role of the CaMKII-β isoform in more relevant cell types.

The main experimental issue to be addressed is as follows. While the reviewers agree that the most interesting outcome of your analysis is the influence of the length of the catalytic domain-hub linker on the nature of the Ca^2+^/CaM-activated autophosphorylation events, they indicate that some further experiments with additional linker lengths would strengthen your conclusions. Currently, you only tested the natural ~30 and ~200 residues. However, it would be of mechanistic interest to know whether the 30 residue linker (in CaMKII-α) is the optimal length for Thr286 trans-autophosphorylation, or whether the optimal length is significantly shorter or longer than this. Likewise, what is the minimal linker length for inhibitory autophosphorylation? Testing additional, intermediate linker lengths would enable you to address this – perhaps natural CaMKII isoform linkers of different lengths exist that could be tested. Also, for completeness sake, it would be ideal if you tested the reciprocal CaMKII-α/β chimera, i.e., CaMKII-α*, in which the CaMKII-α linker is swapped into CaMKII-β.

Another concern is that the model still represents a rather artificial system of ectopically expressed fusion proteins. In particular, the λ phosphatase used here is known to have very broad specificity, suggesting that more relevant and selective Ser/Thr phosphatases that are known to interact and regulate CaMKII, such as PP1 and PP2A, may produce different dephosphorylation kinetics. λ phosphatase is a rather small, highly active protein of ~24 kDa, whereas the eukaryotic cell PPI and PP2A phosphatase holoenzyme complexes are significantly larger, and it is certainly possible that the kinetics of CaMKII dephosphorylation by these enzymes might be significantly slower, perhaps because of steric access issues.

Reviewer #1:

As stated by Bhattancharyya et al. CaMKII is unusual among protein kinases because it is assembled into a large holoenzyme with twelve to fourteen subunits. Analysis of the phosphorylation status of the enzyme using bulk assays, such as Western blots, has been challenging due to difficulties in purifying the holoenzymes, particularly for isoforms with longer kinase-hub linkers. To circumvent this problem the authors designed an elegant designed a single-molecule TIRF microscopy assay that relies on the capture from mammalian cell lysates of CaMKII holoenzymes tagged with fluorescent proteins. Using this approach they conclude that CaMKII-a and CaMKII-b, the two major isoforms in the brain, differ in the balance of activating and inhibitory phosphorylation. This can be explained in part by the differences in the length of a flexible linker that differs between the CaMKII-a and CaMKII-b isoforms.

Overall, this is a well written and detailed study. The rigorous biochemical approach of the Kuriyan lab is balanced with the power of single molecule fluorescence approaches that have been pioneered by the Grove lab. This has led to the formulation of an intricate model for the activation and inactivation of CaMKII's that may prove challenging for some readers to fully appreciate. I only have a few minor comments.

1) I wonder if there is merit in incorporating Figure S3 into the body of the text. I recognize that this is already a vast study of nine full figures, but the data on dephosphorylation in a cellular context is a nice addition to the study.

2) Figure 5 is necessary but quite complicated. I wonder if there is any way to simplify this section.

3) Experiments with small molecule inhibitors such as staurosporine are appropriate, but I wonder if there would be benefit from including work with more established CaMKII antagonists such as KN62.

Reviewer #2:

Bhattacharyya et al. studied the autophosphorylation kinetics of the two main Ca/calmodulin-dependent protein kinase II (CaMKII) isozymes expressed in brain: CaMKII-α and CaMKII-β. To do this, the authors developed a single-molecule assay to measure the phosphorylation status of CaMKII using expression constructs of CaMKII isoforms expressed with a N-terminal fluorescent proteins (mEGFP) fused to a biotin tag for easy capturer on coated glass plates and subsequent visualization, by total internal reflection fluorescence (TIRF) microscopy. Phosphorylation states were detected by phosphorylation site specific antibodies recognizing the activating phosphorylation site Thr286 as well as the two inhibitory sites Thr305/306.

CaMKII constitute a family of enzymes that play a key role in learning and memory by mediating signalling after Ca^2+^ influx. Kinases of this family form dodecameric or tetradecameric holoenzymes that may contain also contain heteroligomeric assemblies of different isoforms that share high sequence homology. The oligomeric state facilitates auto-phosphorylation within the same oligomer allowing for rapid amplification of transient Ca signals. The authors make the intriguing discovery that the linker region between the oligomerization domain and the kinase domain which is much longer in the β isoform modulates autophosphorylation behaviour: CaMKII-α which has a 31-residue linker quickly autophosphorylated on the activating site (Thr286) with relatively little inhibitory phosphorylation on Thr305/306. In contrast, CaMKII-β, which has a longer ~200-residue linker preferentially autophosphorylated at the inhibitory sites Thr305/306 with less activating phosphorylation on the activating site Thr286. A mutant of CaMKII-α that contains the β linker confirmed that the difference in autophosphorylation is indeed a result of the linker region and not due to differences in the catalytic or oligomerization domains. Thus, CaMKII activity in cell can potentially regulated by different ration of α and β isoforms, generating heterooligomers with variable response to activating Ca/calmodulin signals. In addition, de-phosphorylation was studied using non-specific l-phosphatase showing again linker dependent differences.

The reported activation and inactivation data of this key enzyme system are certainly interesting. Experimental data are sound and have been interpreted with care. My only concern is, that the model still represents a very artificial system of ectopically expressed fusion proteins. In particular, the phosphatase used is known to have very broad specificity suggesting that more relevant and selective Ser/Thr phosphatases that are known to interact and regulate CaMKII such as PP1 and PP2a, may produce different dephosphorylation kinetics. However, I think that the in vitro data are however sufficiently interesting for publication in eLife when limitations are more clearly highlighted and they may stimulate more research focussing on the role of the β isoform in more relevant cell types.

Specifically:

a) The notion that α and β ratios regulate activity could be supported by (available) expression data in different cell types. Such data would at least demonstrate that the proposed regulatory mechanism is indeed observed.

b) The heading "The balance between inhibitory and activating autophosphorylation is maintained upon activation of CaMKII in cells" is misleading. The experiments are carried out in lysates using ectopically expressed protein in HEK293T cells. Similarly, the heading "Slower dephosphorylation at the activating site primes CaMKII for activation by incoming Ca^2+^ pulses" suggests that signalling events have been studied downstream of Ca^2+^ pulses – this is not the case as only dephosphorylation kinetics by l-phosphatase has been experimentally assessed. The authors should also discuss that CaMKII activation in neurons is extremely transient (approximately 1 min duration). This is probably not within the time resolution of the current experimental setup.

Reviewer #3:

This biophysical study explores the differences in the autophosphorylation behavior of CaMKII-α and CaMKII-β. These two isoforms differ mainly in the length of the flexible linker between the N-terminal kinase domain and the C-terminal hub domain: ~30 residues for CaMKII-α and ~200 residues for -β. The three autophosphorylation sites examined were Thr286 and Thr305/306, just C-terminal to the kinase domain. It had been shown previously that Thr286 is autophosphorylated in trans (but within the oligomeric CaMKII structure) and activates CaMKII kinase activity, and that Thr305/306 can be autophosphorylated in cis or in trans and is inhibitory. Using single-molecule TIRF microscopy, the authors' main finding is that autophosphorylation of Thr286 is dependent on the length of the kinase-hub linker, with a shorter linker (as in CaMKII-α) facilitating Thr286 autophosphorylation (in trans). They also explored the susceptibility of these phosphorylation sites to phosphatase activity.

Although these results will be of interest to those in the CaMKII field, it is debatable whether the study, in its current form, is of high enough significance to merit publication in *eLife*. Given the authors' main finding that the kinase-hub linker length plays a crucial role in the activating/inhibitory phosphorylation ratio, one would think that they would have explored this feature more exhaustively, for example, by testing intermediate linker lengths (only two tested – the natural ~30 and ~200 residues). Is 30 residues (in CaMKII-α) the optimal length for Thr286 trans-autophosphorylation, or is the optimal length shorter or longer? Also, for completeness sake, the authors should test the reciprocal CaMKII-α/β chimera, i.e., CaMKII-α*, in which the CaMKII-α linker is swapped into CaMKII-β.

---

## [Author Response]

The main experimental issue to be addressed is as follows. While the reviewers agree that the most interesting outcome of your analysis is the influence of the length of the catalytic domain-hub linker on the nature of the Ca^2+^/CaM-activated autophosphorylation events, they indicate that some further experiments with additional linker lengths would strengthen your conclusions. Currently, you only tested the natural ~30 and ~200 residues. However, it would be of mechanistic interest to know whether the 30 residue linker (in CaMKII-α) is the optimal length for Thr286 trans-autophosphorylation, or whether the optimal length is significantly shorter or longer than this. Likewise, what is the minimal linker length for inhibitory autophosphorylation? Testing additional, intermediate linker lengths would enable you to address this – perhaps natural CaMKII isoform linkers of different lengths exist that could be tested. Also, for completeness sake, it would be ideal if you tested the reciprocal CaMKII-α/β chimera, i.e., CaMKII-α*, in which the CaMKII-α linker is swapped into CaMKII-β.

We thank the reviewers for these suggestions. Using five new constructs, we have now studied the effect of kinase-hub linkers of different length on autophosphorylation outcomes in CaMKII. These results are now discussed in the manuscript (subsection CaMKII-β, with a long kinase-hub linker, acquires inhibitory autophosphorylation more readily than CaMKII-α) and included as new supplementary figures (Figure 3—figure supplement 2 and Figure 3—figure supplement 3(A-C)). We believe that the inclusion of these results has strengthened our conclusions. A brief summary of these experimental results is presented below.

1) We have now replaced the linker in CaMKII-αwith that from two other CaMKII isoforms (γwith a 110-residue linker and δwith a 31-residue linker), producing surrogate constructs, CaMKII-γ* and CaMKII-δ*. The autophosphorylation outcomes for shorter-linker CaMKII-δ* is similar to that seen for CaMKII-α, in terms of both the activating and inhibitory phosphorylation (Figure 3—figure supplement 2). The longer-linker CaMKII-γ* exhibits autophosphorylation status that is closer to CaMKII-β/β*, with an increased bias towards inhibitory phosphorylation (Figure 3—figure supplement 2). These results suggest that the longer-linker variants have an increased tendency to autophosphorylate on the inhibitory site, whereas short-linker variants are biased towards activating phosphorylation. The sequences of the linkers in the four CaMKII isoforms are very different, and so these results support the idea that the linker length is an important determinant of phosphorylation outcome.

2) This conclusion was further reinforced by studying a naturally occurring splice-variant of CaMKII-β(CaMKII-β’E) that has a 55-residue linker, instead of the more common ~200 residue linker. We also designed a construct where this truncated linker from CaMKII-β’E replaced the linker in CaMKII-α(CaMKII-β’E*). For both these constructs, we see a large reduction in inhibitory phosphorylation and a small increase in activating phosphorylation when compared to normal CaMKII-β(Figure 3—figure supplement 3A-B).

3) As an additional control, we designed a construct of CaMKII-βin which the kinase-hub linker was replaced by that of CaMKII-α(CaMKII-α*). The autophosphorylation outcomes for CaMKII-α* is similar to that for CaMKII-α (Figure 3—figure supplement 3C).

4) We also made a construct where the entire linker in CaMKII-α is deleted (CaMKII-α-nolinker). We observe that the activating phosphorylation is reduced in this construct when compared to CaMKII-α, while the degree of pThr 305/306 remains unchanged. The deletion of the entire linker may have some steric effects that may influence the autophosphorylation outcomes, and the behavior of this construct requires more study. We have chosen not to include data for the no-linker construct in this manuscript.

We agree that it would be interesting to do a more extensive study of the effects of linker length on phosphorylation outcomes, but note that this will require much more extensive experimentation than is feasible in a short time. Not only would the linker lengths have to be tested, but the sequences of the linkers would also have to be varied.

Another concern is that the model still represents a rather artificial system of ectopically expressed fusion proteins. In particular, the λ phosphatase used here is known to have very broad specificity, suggesting that more relevant and selective Ser/Thr phosphatases that are known to interact and regulate CaMKII, such as PP1 and PP2A, may produce different dephosphorylation kinetics. λ phosphatase is a rather small, highly active protein of ~24 kDa, whereas the eukaryotic cell PPI and PP2A phosphatase holoenzyme complexes are significantly larger, and it is certainly possible that the kinetics of CaMKII dephosphorylation by these enzymes might be significantly slower, perhaps because of steric access issues.

We thank the reviewers for raising this point. We note that we have reported dephosphorylation kinetics using purified PP1αcatalytic subunit (Figure 7—figure supplement 1B-C). We do recognize that the PP1 and PP2A phosphatases in the eukaryotic cell act in concert with other regulatory subunits, and that these regulatory subunits are not included in our study. A detailed investigation of the effect of phosphatase holoenzymes is beyond the scope of the current manuscript.

Reviewer #1:[…]Overall, this is a well written and detailed study. The rigorous biochemical approach of the Kuriyan lab is balanced with the power of single molecule fluorescence approaches that have been pioneered by the Grove lab. This has led to the formulation of an intricate model for the activation and inactivation of CaMKII's that may prove challenging for some readers to fully appreciate. I only have a few minor comments.1) I wonder if there is merit in incorporating Figure S3 into the body of the text. I recognize that this is already a vast study of nine full figures, but the data on dephosphorylation in a cellular context is a nice addition to the study.

Thank you for this suggestion. We have now included the data on phosphorylation and dephosphorylation in the cellular context as part of a main figure (Figure 3C-D).

2) Figure 5 is necessary but quite complicated. I wonder if there is any way to simplify this section.

Thank you for this suggestion as well. We have now revised Figure 5 to highlight only the key reaction pathways for clarity. In the figure legend, we have mentioned that this is a simplified schematic of our kinetic model and a detailed description of all the reactions is provided in the Appendix.

3) Experiments with small molecule inhibitors such as staurosporine are appropriate, but I wonder if there would be benefit from including work with more established CaMKII antagonists such as KN62.

We thank the reviewer for this suggestion. In the current experiment, staurosporine is used to completely abolish CaMKII kinase activity, which is achieved at the high staurosporine concentrations (100 μM) that were used. The same level of kinase inhibition can most likely be achieved at a much lower concentration with the more established CaMKII antagonists and we will integrate this idea in future investigations.

Reviewer #2:[…]The reported activation and inactivation data of this key enzyme system are certainly interesting. Experimental data are sound and have been interpreted with care. My only concern is, that the model still represents a very artificial system of ectopically expressed fusion proteins. In particular, the phosphatase used is known to have very broad specificity suggesting that more relevant and selective Ser/Thr phosphatases that are known to interact and regulate CaMKII such as PP1 and PP2a, may produce different dephosphorylation kinetics. However, I think that the in vitro data are however sufficiently interesting for publication in eLife when limitations are more clearly highlighted and they may stimulate more research focussing on the role of the β isoform in more relevant cell types.

We thank reviewer 2 for the positive feedback on our work. We agree that our conclusions are based on in vitroexperiments on ectopically expressed fusion proteins. Future investigations will focus on expanding the scope of our current findings to more relevant cell types and understand the physiological relevance of different CaMKII isoforms.

We agree with the reviewer regarding the choice of phosphatase. We note that we have reported dephosphorylation kinetics using purified PP1 catalytic subunit (Figure 7—figure supplement 1B-C). We do recognize that the PP1 and PP2A phosphatases in the eukaryotic cell act in concert with other regulatory subunits, and that these regulatory subunits are not included in our study. A detailed investigation of the effect of phosphatase holoenzymes is beyond the scope of the current manuscript.

Specifically:a) The notion that α and β ratios regulate activity could be supported by (available) expression data in different cell types. Such data would at least demonstrate that the proposed regulatory mechanism is indeed observed.

Thank you for this suggestion. In the concluding remarks, we have now clarified that the expression levels of CaMKII-αand CaMKII-βin neurons is dynamic and is altered by synaptic activity (Thiagarajan et al., 2002). High neuronal activity increases the α/βratio and *vice versa*, resulting in functional differences between these isoforms. This difference in the expression level between two isoforms can manifest as differences in cellular targeting, stimulus frequency response, and as we now show, can switch the balance between activating and inhibitory autophosphorylation.

b) The heading "The balance between inhibitory and activating autophosphorylation is maintained upon activation of CaMKII in cells" is misleading. The experiments are carried out in lysates using ectopically expressed protein in HEK293T cells.

In these experiments, we are activating CaMKII variants in HEK293T cells using ionomycin and Ca^2+^. After activation, the cells are lysed, the activated holoenzymes are immobilized on the coverslip and their autophosphorylation status is directly measured without any further manipulation to the protein. To avoid confusion, we have now modified the heading slightly to say “The balance between inhibitory and activating autophosphorylation is maintained upon activation of CaMKII in HEK293T cells”. In addition, the Materials and methods section is also updated to explain this experiment more clearly.

Similarly, the heading "Slower dephosphorylation at the activating site primes CaMKII for activation by incoming Ca^2+^ pulses" suggests that signalling events have been studied downstream of Ca^2+^ pulses – this is not the case as only dephosphorylation kinetics by l-phosphatase has been experimentally assessed.

We agree with the reviewer that this heading is misleading as the signaling events downstream of the Ca^2+^ pulses have not been studied. In the revised version, we have modified the heading to say “Slower dephosphorylation at the activating site primes CaMKII for activation by subthreshold concentrations of Ca^2+^/CaM”.

The authors should also discuss that CaMKII activation in neurons is extremely transient (approximately 1 min duration). This is probably not within the time resolution of the current experimental setup.

We have addressed this point in the concluding remarks (highlighted in yellow). The observation of transient activation of CaMKII in neurons relies on the use of a FRET-based sensor, Camui. Camui reports on a conformational change undergone by CaMKII during activation, and does not measure the phosphorylation states directly.

Reviewer #3:[…]Although these results will be of interest to those in the CaMKII field, it is debatable whether the study, in its current form, is of high enough significance to merit publication in eLife. Given the authors' main finding that the kinase-hub linker length plays a crucial role in the activating/inhibitory phosphorylation ratio, one would think that they would have explored this feature more exhaustively, for example, by testing intermediate linker lengths (only two tested – the natural ~30 and ~200 residues). Is 30 residues (in CaMKII-α) the optimal length for Thr286 trans-autophosphorylation, or is the optimal length shorter or longer? Also, for completeness sake, the authors should test the reciprocal CaMKII-α/β chimera, i.e., CaMKII-α*, in which the CaMKII-α linker is swapped into CaMKII-β.

We thank reviewer 3 for the feedback, especially highlighting that an analysis connecting the kinase-hub linker length to autophosphorylation outcomes in CaMKII is missing. In response, we have now added experimental results from five new constructs in the revised manuscript, as summarized above. We believe that the inclusion of these data in the manuscript (subsection “CaMKII-β, with a long kinase-hub linker, acquires inhibitory autophosphorylation more readily than CaMKII-α”, Figure 3—figure supplement 2 and Figure 3—figure supplement 3A-C) has strengthened the conclusions of our work. A summary of these experimental results is also presented below.

1) We have now replaced the linker in CaMKII-αwith that from two other CaMKII isoforms (γwith a 110-residue linker and δwith a 31-residue linker), producing surrogate constructs, CaMKII-γ* and CaMKII-δ*. The autophosphorylation outcomes for shorter-linker CaMKII-δ* is similar to that seen for CaMKII-α, in terms of both the activating and inhibitory phosphorylation (Figure 3—figure supplement 2). The longer-linker CaMKII-γ* exhibits autophosphorylation status that is closer to CaMKII-β/β*, with an increased bias towards inhibitory phosphorylation (Figure 3—figure supplement 2). These results suggest that the longer-linker variants have an increased tendency to autophosphorylate on the inhibitory site, whereas short-linker variants are biased towards activating phosphorylation. The sequences of the linkers in the four CaMKII isoforms are very different, and so these results support the idea that the linker length is an important determinant of phosphorylation outcome.

2) This conclusion was further reinforced by studying a naturally occurring splice-variant of CaMKII-β(CaMKII-β’E) that has a 55-residue linker, instead of the more common ~200 residue linker. We also designed a construct where this truncated linker from CaMKII-β’E replaced the linker in CaMKII-α(CaMKII-β’E*). For both these constructs, we see a large reduction in inhibitory phosphorylation and a small increase in activating phosphorylation when compared to normal CaMKII-β(Figure 3—figure supplement 3A-B).

3) As an additional control, we designed a construct of CaMKII-βin which the kinase-hub linker was replaced by that of CaMKII-α(CaMKII-α*). The autophosphorylation outcomes for CaMKII-α* is similar to that for CaMKII-α (Figure 3—figure supplement 3C).

4) We also made a construct where the entire linker in CaMKII-α is deleted (CaMKII-α-nolinker). We observe that the activating phosphorylation is reduced in this construct when compared to CaMKII-α, while the degree of pThr 305/306 remains unchanged. The deletion of the entire linker may have some steric effects that may influence the autophosphorylation outcomes, and the behavior of this construct requires more study. We have chosen not to include data for the no-linker construct in this manuscript.

We agree that it would be interesting to do a more extensive study of the effects of linker length on phosphorylation outcomes, but note that this will require much more extensive experimentation than is feasible in a short time. Not only would the linker lengths have to be tested, but the sequences of the linkers would also have to be varied.